



# Characteristics of ecosystems under various anthropogenic impacts in a tropical forest region of Southeast Asia

Chansopheaktra Sovann[1,2], Torbern Tagesson[1], Patrik Vestin[1], Sakada Sakhoeun[3], Soben Kim[4], Sothea Kok[2], Stefan Olin[1]

[1]Department of Physical Geography and Ecosystem Science, Lund University, Sölvegatan 12, S-223 62 Lund, Sweden
[2]Department of Environmental Science, Royal University of Phnom Penh, Phnom Penh, 120404, Cambodia
[3]Provincial Department of Environment, Ministry of Environment, Siem Reap, 171202, Cambodia
[4]Faculty of Forestry, Royal University of Agriculture, Phnom Penh, 120501, Cambodia

*Correspondence to*: Chansopheaktra Sovann (chansopheaktra.sovann@nateko.lu.se)

**Abstract.** Given the severe anthropogenic pressure on tropical forests and the high demand for field observations of ecosystem characteristics, it is crucial to collect such data both in pristine tropical forests and in the converted deforested land-cover classes. To gain insight into the ecosystem characteristics of pristine tropical forests, regrowth forests, and cashew plantations, we established an ecosystem monitoring site in Phnom Kulen National Park, Cambodia. Here, we present the first observed datasets at this site of forest inventories, leaf area index, leaf traits of woody species, a fraction of intercepted photosynthetically active radiation, and soil and meteorological conditions. We examined how land-cover change affects the species and functional diversity, stand structure, and soil conditions among the three land-cover classes. We found significant reductions in several ecosystem characteristics, caused by the anthropogenic land cover conversion, which underlines the profound impact land-cover change has on ecosystem productivity, resilience, and functioning in these tropical forest regions. We further investigated relationships between diameters at breast height and tree height, and demonstrated the feasibility of locally updating aboveground biomass estimates using power-law functions. These datasets and findings can contribute to filling data gaps in tropical forest research, addressing global environmental challenges, and supporting sustainable forest management. The datasets are available at https://doi.org/10.5281/zenodo.10146582 (Sovann et al., 2024a) and https://doi.org/10.5281/zenodo.10159726 (Sovann et al., 2024b), and future data from the field site will be uploaded on a regular basis to https://zenodo.org/communities/cambodia_ecosystem_data.

**Keywords:** tropical forest, forest ecosystem, forest inventory, biomass, Kulen, Cambodia.



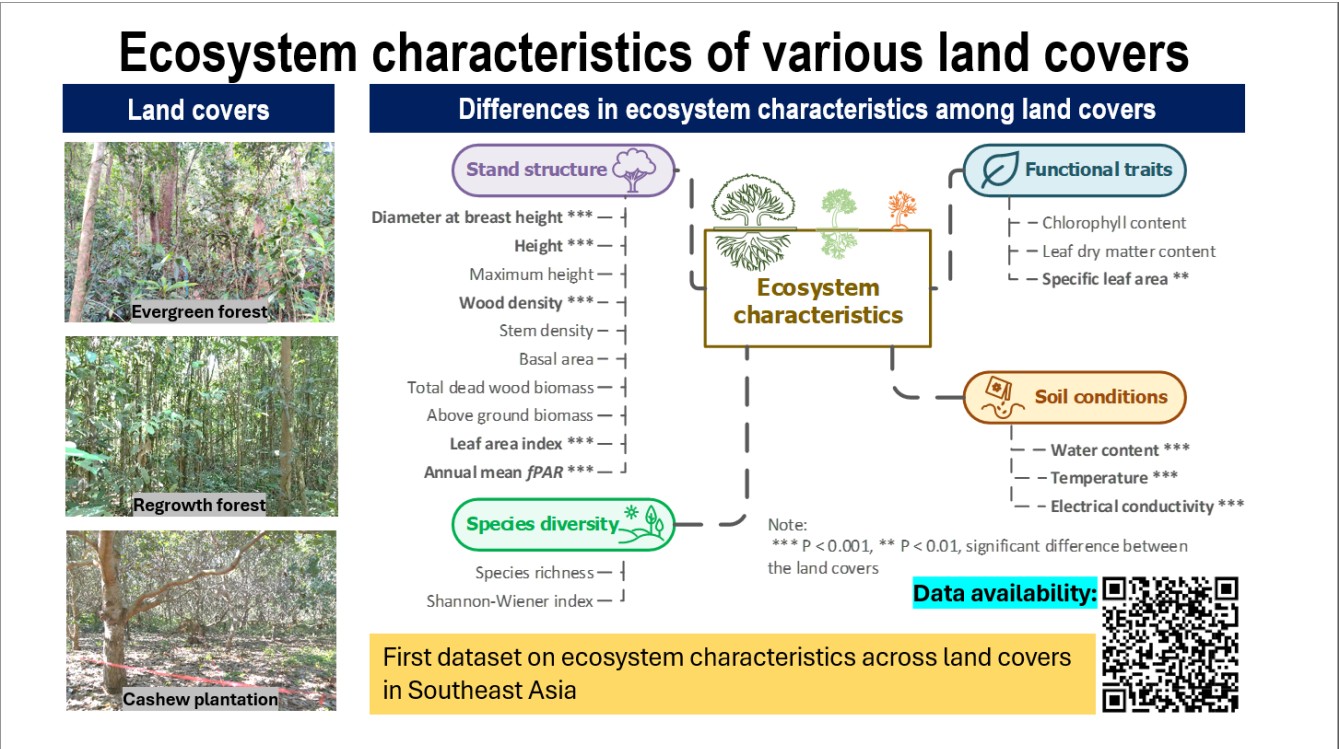

## 1 Introduction

Tropical forests cover approximately 14 % of the Earth's surface (Fichtner and Härdtle, 2021) and contribute significantly to global terrestrial biodiversity (Giam, 2017) and biogeochemical cycles (Artaxo et al., 2022). Tropical forests produce at least 30 % of the global terrestrial net primary production (Townsend et al., 2011; Joseph Wright, 2013) and store approximately 60 % of the global terrestrial biomass (Pan et al., 2013). In addition, they play a critical role in regulating hydrological cycles on a continental scale (Gloor et al., 2013). Tropical forests have been under severe anthropogenic pressures from agricultural land expansion, resource exploitation (logging, mining), and urbanization (Gardner et al., 2009; Laurance et al., 2013). Due to these disturbances, tropical forest ecosystems have degraded, resulting in a decrease in biodiversity (Barlow et al., 2016). Southeast Asia, though harbouring roughly 15 % of the world's tropical forests (Stibig et al., 2014), has suffered the highest global deforestation rates over the past 15 years (Miettinen et al., 2011). This alarming trend threatens over 40 % of the region's biodiversity by 2100 (Sodhi et al., 2004). The forests are mainly disturbed by timber harvesting (Pearson et al., 2017), slash-and-burn agriculture, and agricultural plantations as a consequence of fulfilling global demands for timber production and agricultural commodities, especially rubber, cashew, oil palm, Eucalyptus and Acacia (Phompila et al., 2014; Grogan et al., 2015; Chen et al., 2016; Johansson et al., 2020). In addition to primary forests, secondary forests that regenerate after clear-cutting or other ecosystem disturbances are also important for protecting biodiversity and assuring the availability of ecosystem





services and goods (Tito et al., 2022). However, despite the significance of tropical forests in biodiversity conservation and

ecosystem services, little is known about how the conversion from primary to secondary forests and plantations impacts biodiversity and ecosystem functioning (Edwards et al., 2011; Singh et al., 2014).

In the context of tackling the current global environmental challenges, field observation data are necessary to assess the dynamic responses of ecosystems to changing environmental conditions on fine spatial and temporal scales. Field observations of ecosystem characteristics provide crucial insights into key ecosystem functions and services, such as vegetation

productivity, carbon sequestration, hydrological cycle, ecosystem stability and resilience to disturbances, nutrient reservoir capacity, and the abundance of habitats of organisms (Naeem et al., 1994; Hector, 1998; Cardinale et al., 2012; Chen et al., 2016; Liang et al., 2016; Parisi et al., 2018b; Woodall et al., 2020). Since the collection of field observations is time consuming and labour intensive, the availability of such data is in general limited, and in particular from tropical forests (DeFries et al., 2007; Li et al., 2021). In addition, field data are important for the parameterization and evaluation of remote sensing products

and dynamic vegetation models, essential for modelling and upscaling ecosystem responses to anthropogenic disturbances and climate change (Feng et al., 2018). Hence, there is an increasing demand for field observations of environmental variables and ecosystem characteristics, resulting in open data repositories such as FLUXNET, ICOS Carbon Portal, SpecNet, and the TRY database (Gamon et al., 2010; Kattge et al., 2020; Pastorello et al., 2020).

Given the severe anthropogenic pressures on tropical forests and the high demand for field observations of ecosystem

characteristics of tropical forests, it is crucial to collect such data, both in pristine tropical forests and in the land cover the deforested regions are converted into. The data from different land-cover classes will aid in quantifying the effect the land-cover conversion has on the ecosystem characteristics. To gain insight into the impact of land cover conversion on ecosystem characteristics, we therefore established an ecosystem monitoring site in Phnom Kulen National Park (Kulen), Cambodia. The main aim of this study is to describe the first unique in situ data of ecosystem characteristics of pristine tropical forests,

regrowth forests, and cashew plantations from this novel monitoring site. The ecosystem level collected data include [1] forest inventory, [2] leaf traits of woody species, [3] leaf area index (*LAI*), [4] fraction of photosynthetically active radiation (*fPAR*), and [5] soil conditions. We also installed a weather station to collect data of the meteorological conditions of the wider area (the Kulen National Park). With these data, we will analyse the difference in ecosystem characteristics of these land-cover classes, explore relationships between ecosystem characteristics, and how these are affected by conversions between these

land-cover classes.

## 2 Materials and Methods

### 2.1 Study area and selection of plots

The selected study area is the Phnom Kulen National Park located in the Siem Reap Province in north-west Cambodia (Fig. 1). It covers 37,380 ha predominantly on sandstone plateaus with the highest peak of 496 m (Geissler et al., 2019). Kulen is a

hotspot for ecosystem service provisioning in Cambodia, mainly for water supply, potential carbon sink, and cultural services





(Jacobson et al., 2022). It is the origin of the Khmer Empire and contains numerous archaeological sites. The stream water from the mountain is not only used to support local livelihoods in water supply and irrigation downstream (Somaly et al., 2020). It is also the primary water source to recharge surface water and groundwater aquifers in the Angkor Wat, UNESCO World Heritage Site. Hence, the area is of high importance to ensure that the temples' foundations remain stable and maintain

their surrounding forest ecosystem (Peou et al., 2016). However, previous studies revealed that the forestland in and around Kulen has been disturbed (Chim et al., 2019). The three main land-cover classes on Kulen are [1] nearly intact tropical evergreen forests (EF), [2] forests that regrow naturally after clear-cutting (RF) and [3] household-scale cashew plantations (CP). Approximately 60 % of Kulen is today covered by cashew plantations, another 13 % consists of forestland, while the remainder comprises other land-cover classes (Singh et al., 2019).


**Figure 1. The locations of the nine forest inventory plots and the meteorological station in the Phnom Kulen National Park, Cambodia. Note: the background land use 2021 was derived from SERVIR-Mekong (2024).**

Nine forest inventory plots were established in Kulen in December 2020, three within each of the EF, RF, and CP land-cover classes (Fig. 1), with a minimum separation of 250 meters to capture stand structure variation for each land-cover class. The



EF plots represented tropical evergreen forests with no clear-cut history. The RF plots were dominated by at least 10 year old natural regrowth forests, RF1 was clear-cut in 2009, while RF2 and RF3 experienced timber harvesting and burning in 2006, with human disturbances continuing until 2013. The CP plots were permanent rainfed cashew plantations, with cashew trees planted in 2013 in CP1 and in 2012 for the other two. For additional plot characteristics, including photographs, locations, elevations, slopes, meteorological conditions, soil types, and geological types, see Table S1.1 and Fig. S1.1.

## 2.2 Data collection

### 2.2.1 Forest inventory

The forest inventory was performed by following the standard method of the National Forest Inventory of Cambodia (Than et al., 2018). Each plot was designed as a rectangle with 50 m x 30 m long edges in the south-north and west-east directions. The plots were further subdivided into five subplots with the following dimensions: 2 m x 2 m, 5 m x 5 m, 10 m x 10 m, 30 m x 15 m, and 30 m x 50 m (Fig. S1.2). In the 2 m × 2 m subplots, seedlings with diameters at breast height (*DBH*, 1.3 m above ground) of less than 1 cm were recorded. In the 5 m x 5 m, 10 m x 10 m, 30 m x 15 m, and 30 m x 50 m subplots, trees with *DBH* ranges of 1–5 cm, 5–15 cm, 15–30 cm, and greater than 30 cm were measured, respectively.

For seedlings, we only recorded the total numbers of each species. For the *DBH* range of 1–5 cm, we noted the *DBH*, tree height (*H*), species, local name (Khmer), and position of each tree. For trees with a *DBH* greater than 5 cm, we collected the same data as for trees with a *DBH* of 1–5 cm, plus bole height (the height from the ground to the first main lowest stem), health (healthy or infected), quality (straight, bent, or crooked stem), origin (natural or planted), and stump diameter and height (measured 15 cm above ground for annual tree growth monitoring).

Deadwood is a significant indicator of decomposition and nutrient cycling processes in a forest ecosystem (Shannon et al., 2022). Data on lying and standing deadwood with a *DBH* greater than 10 cm in the 30 m x 15 m subplots were also collected. The deadwood decomposition levels were classified into five scales, based on harmonizing the scaling systems of the National Forest Inventory of Sweden (Swedish NFI, 2019) and Cambodia (Than et al., 2018) (Table S1.2). For standing deadwood, we recorded their species, local name, location, height, and decomposition level. For lying deadwood, we counted the number of pieces and measured their lengths, base and tree diameters, and decomposition levels.

### 2.2.2 Leaf sample collection and measurement

A total of 453 leaf samples from 30 woody species were collected inside and 500 m around the forest inventory plots in December 2019 and August 2022. Each species was represented by five to 47 leaf samples. Each leaf's fresh mass, chlorophyll content, and photo were taken in the field. A Chlorophyll Meter (SPAD 502 Plus; Konica Minolta Sensing Inc., Japan) was used in situ to measure chlorophyll content five times on each leaf surface to retrieve a leaf mean value. The given measurement unit was in SPAD value (Soil Plant Analysis Development) and later converted to chlorophyll a and b content (*Chl*) in µg cm$^{-2}$ (Coste et al., 2010). We obtained fresh leaf mass by weighting in the field and leaf dry mass by oven-drying the leaves at 60



°C until the leaf mass remained constant (oven-dried for at least three days) (Garnier et al., 2001). The leaf photos were used for estimating leaf lengths and areas using ImageJ (Schindelin et al., 2012; Schneider et al., 2012).

### 2.2.3 Meteorological and photosynthetically active radiation data

A meteorological station was installed in an open area to continuously record metrological conditions, and incoming
photosynthetically active radiation (*PAR*) for the wider area (the Kulen National Park). Data were sampled at one minute intervals and stored as 15 minute averages (sum for rainfall). The installation was done in November 2020 in Khnang Phnom Commune, Svay Luer District, Siem Reap Province, at 13° 34' 16.1148" N, 104° 9' 45.6768" E, and an altitude of 314 m above mean sea level. The station has one Atmos 41 meteorological station (Meter Group Inc. WA, USA), installed 2.2 m above ground level, measuring rainfall, wind speed, wind direction, global radiation, atmospheric pressure, and air temperature.
Additionally, four *PAR* sensors (SQ-110-SS, Apogee Instruments, Inc., UT, USA) were positioned 2 m above the ground to record incoming *PAR* ($PAR_{inc}$) (Fig. S2.1).

Six additional loggers with five *PAR* sensors (SQ-521-SS and SQ-110-SS, Apogee Instruments, Inc., UT, USA) and one TEROS 12 soil moisture sensor each (Meter group Inc. WA, USA), collecting data at a 15 minute mean timestep, were installed in six of the forest inventory plots in April 2022. The soil moisture sensors were installed at a depth of 20 cm to measure soil
water content (*SWC*), soil temperature (*Ts*), and soil electrical conductivity (*ECs*). Two loggers were placed in each land-cover class (EF, RF, and CP). The selection of plots in each land-cover class was based on previous measurements of leaf area index (*LAI*) and the loggers were placed at the plots with the highest and lowest *LAI* for each land cover, respectively. Thus, the selected plots for installing *PAR* sensors were EF1, EF3, RF1, RF3, CP2, and CP3 (Fig. 1). The *PAR* sensors were placed with one in the centre of the plot and the other four placed 15 ± 1 m apart at 30°, 150°, 220°, and 330° from the north. In cases of
unfavourable field conditions, such as high termite nests or being too close to a tree, the locations were adjusted 0.5−1 m east or west of the planned position. Each *PAR* sensor was mounted on 1.3 m poles to record *PAR* below canopy data. We calculated the fraction of *PAR* intercepted by the stand canopy (*fPAR*) for each plot using Eq. (1) (Olofsson and Eklundh, 2007). Each TEROS 12 soil moisture sensor was installed at a depth of 20 cm in the middle of the six plots to measure *SWC*, *Ts*, and *ECs*. The data of *fPAR* and soil conditions from two plots within the same land-cover classes were averaged to represent those
classes.

$$fPAR = \frac{(PAR_{inc} - PAR_{below})}{PAR_{inc}} \tag{1}$$

Where $PAR_{inc}$ and $PAR_{below}$ are photosynthetically active radiation above and below canopy (μmol m$^{-2}$ s$^{-1}$). $fPAR$ is in percentage.





### 2.2.4 Leaf area index measurements

We measured each plot's total one-sided leaf surface area per unit ground area, *LAI*, using a LAI-2000 Plant Canopy Analyzer
(LI-COR, NE, USA). The measurements were conducted six times across two seasons: four times during the dry season
(November/December 2019, November 2020, December 2020, and March 2021) and twice during the rainy season (September
2020 and June 2021). The measurements were taken both at ground level to capture the total *LAI* ($LAI_T$) and at breast height
to specifically assess tree canopy *LAI* ($LAI_C$) within two diagonal transects across the 50 m x 30 m rectangular plots. On each
measurement occasion, we collected between 32 and 75 samples, except for the ground-level measurements of the RF3 plot
in December 2020, where only ten samples were collected due to technical issues.

### 2.3 Data analysis

#### 2.3.1 Species diversity

We investigated the species diversity of various land covers by calculating species richness ($S_R$) and the Shannon-Wiener index
($S_H$) (Shannon, 1948). The $S_R$ was determined by summing the number of tree species in each plot. The $S_H$ is commonly used
to quantify species richness and evenness in a community by representing the number of species and how equally individuals
are distributed among them (Hill, 1973). The value of $S_H$ increases as the number of species and the degree of evenness
increase. The $S_H$ was calculated by:

$$S_H = - \sum_{i=1}^{n} P_i \ln(P_i) \tag{2}$$

Where $S_H$ is Shannon-Wiener index (unitless), $P_i$ is a proportion of *i* species in a community (unitless), and *n* is the number
of species in a plot (unitless). We calculated the $S_R$ and $S_H$ at the plot level and then averaged the values for each land-cover
class.

#### 2.3.2 Functional traits and diversity

We computed the specific leaf area (*SLA*) for each of the 453 leaf samples as the ratio of leaf area to leaf dry mass. Likewise,
leaf dry matter content (*LDMC*) was calculated by the ratio of dry leaf mass to fresh leaf mass (Garnier et al., 2001; Akram et
al., 2023). We estimated the trait community-weighted means and standard deviations of $SLA_{cwm}$, $LDMC_{cwm}$, and $Chl_{cwm}$ to
represent ecosystem functions and their diversity at the land-cover level (Garnier et al., 2004; Leoni et al., 2009; Wang et al.,
2020) with:

$$T_{cwm} = \frac{\sum_{i=1}^{n} W_i T_i}{\sum_{i=1}^{n} W_i} \tag{3}$$



Where $T_{\mathrm{cwm}}$ is trait community-weighted mean for *SLA*, *LDMC*, or *Chl*, $T_i$ is the species-specific trait value tree $i$, $n$ is total number of trees, $W_i$ is the weight (volume based) value of the tree, assuming that larger trees have a greater impact on the ecosystem function (Chave et al., 2005; Feldpausch et al., 2011). Before computing $T_{\mathrm{cwm}}$ for each trait, we addressed missing species traits within each plot by first taking values from a different plot with the same land-cover class. If unavailable, we sought values from the same species across all nine plots. If neither was available, we used the mean trait value of the plot.

### 2.3.3 Stand structural attributes

We examined the differences in *DBH*, *H*, basal area (*BA*), aboveground biomass (*AGB*), and deadwood biomass (*DWB*) for the various land-cover classes to characterize stand structure attributes. Deadwood volumes ($V_{\mathrm{DW}}$, m$^3$) for each bole were determined by Smalian's equation:

$$V_{DW} = (\pi\, H_{\mathrm{b}})\, \frac{\left(D_{\mathrm{base}}^2 + D_{\mathrm{top}}^2\right)}{8} \tag{4}$$

Where $D_{\mathrm{base}}$ and $D_{\mathrm{top}}$ are diameters at base and top (m), and $H_{\mathrm{b}}$ is the length/height of the trunk (m).

Deadwood biomass was then received by multiplying $V_{\mathrm{DW}}$ with a mean deadwood density of 0.45 g cm$^{-3}$ (Kiyono et al., 2007). Total *DWB* was computed plot-wise by taking the sum of lying and standing *DWB*. *DWB* for each land-cover class was calculated as the average of the total *DWB* across the plots within that land-cover class.

Basal area was determined plot-wise by combining the *DBH* of all living trees within a plot:

$$BA = \sum_{i=1}^{n} \pi \left(\frac{DBH_i}{2}\right)^2 \left(\frac{10^4}{A_i}\right) \tag{5}$$

Where *BA* is a plot-wise total basal area of all living trees (m$^2$ ha$^{-1}$), $n$ is a number of trees in a plot, $DBH_i$ is the diameter at breast height of tree $i$ in a sampling plot (m), $\pi\left(\frac{DBH_i}{2}\right)^2$ is the circle basal area of tree $i$ (m$^2$), $\left(\frac{10^4}{A_i}\right)$ are the scaling factors employed to convert the sampled subplot area ($A_i$) to one hectare (unitless). The *BA* for each land-cover class was represented by the mean *BA* of all plots within a class.

We calculated the mean and standard deviation of *DBH* and *H* for each plot and land cover. We further used these for establishing relationships between *DBH* and *H*, as such relationships serve as functional traits characterizing tree growth patterns and successional stages within forest communities (Nyirambangutse et al., 2017; Howell et al., 2022). We used natural logarithms and then converted them to power-law relationships both plot- and land-cover class-wise (West and Brown, 2005). An ordinary least-square linear regression (OLS) was applied to investigate the *DBH-H* relationship, followed by transforming the relationship into a power-law relationship (Huxley, 1932).



$$H = K_1 DBH^{K_2} \qquad (6)$$

Where $K_1$ and $K_2$ are the power-law intercept and slope, respectively. The $K_1$ captures the overall scaling relationship between $H$ (m) relative to $DBH$ (cm) within a forest community while $K_2$ regulates the rate of $H$ increase relative to $DBH$ growth.

The obtained $K_1$ and $K_2$ values were further used to estimate $AGB$ ($AGB_h$) Eq. (7) in Table 1. We also computed the $AGB$ using existing equations (Table 1, Eqs. (9–11)) ($AGB_f$) adopted for the three different land-cover classes. These EF and RF allometric

equations were developed for tropical multiple species, whereas the CP was a species-specific allometric equation for the cashew tree ((Malimbwi et al., 2016). The wood density ($WD$) values required for the $AGB$ estimations were species-specific and obtained from The International Council for Research in Agroforestry (2022) and Zanne et al. (2009). When multiple $WD$ values for a tree species were available, the mean value was used, whereas when no species-specific $WD$ values were available, the average of tropical Asia (0.57 g cm$^{-3}$) was used (Reyes et al., 1992). The applied $WD$ values for this study then ranged from

0.39–1.04 g cm$^{-3}$. Specifically, the $WD$ values (mean ± a standard deviation) for EF, RF, and CP were 0.74 ± 0.17 g cm$^{-3}$, 0.72 ± 0.15 g cm$^{-3}$, and 0.45 g cm$^{-3}$, respectively. We first estimated $AGB$ at the plot level in kilograms, then scaled these values to megagrams per hectare, and averaged per land-cover class.

**Table 1. Allometric equations used for estimating aboveground biomass ($AGB$, kg tree$^{-1}$) in the different land-cover classes.**

| No. | Equations | Land cover | $AGB$ allometric equations | Regions | n | $DBH$ (range, cm) | $\overline{WD}_f$ (mean ± SD, g cm$^{-3}$) | References |
|-----|-----------|------------|----------------------------|---------|---|-------------------|--------------------------------------------|------------|
| 1 | Eq. (7) | All | $AGB_h = \dfrac{WD\,\pi\,K_1}{8}\,DBH^{2+K_2} + \varepsilon$ | - | - | - | - | This study |
| 2 | Eq. (8) | All | $AGB_{wd} = \dfrac{WD}{\overline{WD}_f}\,AGB_f$ | - | - | - | - | This study |
| 3 | Eq. (9) | EF | $AGB_f = 0.1184\,DBH^{2.53}$ | Pantropical | 170 | 5.0–148.0 | 0.58 ± 0.02 | Brown (1997) |
| 4 | Eq. (10) | RF | $AGB_f = 0.0829\,DBH^{2.43}$ | Sarawak, Malaysia | 136 | 0.1–28.7 | 0.38 ± 0.07 | Kenzo et al. (2009) |
| 5 | Eq. (11) | CP | $AGB_f = 0.8450\,DBH^{1.77}$ | Pwani, Tanzania | 45 | 6.0–89.9 | 0.18 | Malimbwi et al. (2016) |

**Note: EF is evergreen forests, RF is regrowth forests, CP is cashew plantations. In Eqs. (9–11), $DBH$ is diameter at breast height**
**(cm), and $\overline{WD}_f$ is the reported mean wood density used in $AGB_f$ (kg m$^{-3}$). In Eq. (7), $K_1$ and $K_2$ are derived power-law intercept and slope values between $DBH$ (cm) and tree height ($H$, m) relationship in Eq. (6), $\varepsilon$ is a statistical error term, $WD$ is wood density for each tree species (g cm$^{-3}$), and $DBH$ is in centimetres. In this study, in Eq. (7), we employed a trunk shape factor of 1/8 for calculating the volume of frustum cones, as proposed by King et al. (2006). This factor falls within the range of 1/4 (cylinder volumes) to 1/12 (cone volumes). In Eq. (8), $AGB_{wd}$ is our examined aboveground biomass based on equations Eqs. (9–11) with species-specific wood**
**density updated for our woody tree species, $WD$ are the species-specific wood density of trees in each plot (g cm$^{-3}$).**



### 2.3.4 Statistical analysis

Descriptive statistics were conducted to examine the difference in ecosystem characteristics between plots and land-cover classes. One-way ANOVA tests (ANOVA) were used to assess significant differences in mean values across land-cover classes. Tukey's Honestly Significant Difference test (Tukey HSD) was further employed for pairwise comparisons between
land-cover classes. Pearson correlation and ordinary least squares regression analyses were used to explore relationships between variables. All analyses were performed using R 4.2.3 (R Core Team, 2023).

## 3 Results

### 3.1 Meteorological conditions

The observed annual daily mean air temperature from April 2022 to April 2023 at Kulen meteorological station was 24.2 ±
2.0 °C, varying between 17.8 °C and 28.6 °C (Fig. 2a). The total annual rainfall was 2290 mm, significantly surpassing nearby lowland stations: Banteay Srei station, located 22 km west, recorded 1160 mm, and Siem Reap City station, situated 40 km southwest, recorded 1475 mm (Chim et al., 2021). About 90 % of the annual precipitation fell during the rainy season from May to November, with September being the wettest month (505 mm). The daily maximum rainfall can reach up to 141 mm, but the daily mean during the rainy season was 11.2 ± 19.7 mm (Fig. 2b). The annual daily mean of global radiation, relative
humidity, vapour pressure deficit, and wind speed were $172 \pm 44$ W m$^{-2}$, $88 \pm 12$ %, $0.45 \pm 0.21$ kPa, and $0.68 \pm 0.22$ m s$^{-1}$, respectively (Fig. 2c–f).

For the different land-cover classes, daily mean soil water content ranged between 0.14–0.23 m$^3$ m$^{-3}$, soil temperature between 24.2–25.8 °C, and soil electrical conductivity between 0.025–0.039 dS m$^{-1}$ (measured at 20 cm depth, Fig. 2g–i). In particular, the mean $Ts$ at CP ($25.8 \pm 1.5$ °C) was significantly higher than for EF ($24.3 \pm 1.2$ °C) and RF ($24.2 \pm 1.3$ °C), whereas the
mean $SWC$ was significantly lower in RF ($0.14 \pm 0.03$ m$^3$ m$^{-3}$) compared to EF ($0.23 \pm 0.06$ m$^3$ m$^{-3}$) and CP ($0.21 \pm 0.05$ m$^3$ m$^{-3}$) (Table 2). Additionally, EF had higher $ECs$ (0.039 dS m$^{-1}$) than RF and CP (0.032 dS m$^{-1}$, 0.025 dS m$^{-1}$) (p-value < 0.001), indicating higher salinity levels in the soil.







**Figure 2. The meteorological conditions at Kulen meteorological station (a–f), and soil conditions at each land-cover class (g–i) from April 10, 2022, to April 9, 2023. (a) Daily mean air temperature ($T_{air}$, °C), (b) daily total precipitation ($P$, mm), (c) daily mean global radiation ($Rg$, W m$^{-2}$), (d) daily mean relative humidity ($RH$, %), (e) daily mean vapour pressure deficit ($VPD$, kPa), and (f) daily mean wind speed ($WS$, m s$^{-1}$), (g) daily mean soil water content ($SWC$, m$^3$ m$^{-3}$), (h) daily mean soil temperature ($Ts$, °C), (i) daily mean soil saturation extraction electrical conductivity ($ECs$, dS m$^{-1}$). The vertical dashed line region in all the plots highlighted the rainy season period in Cambodia from May to October. The grey-shaded regions around the mean in (a), (d), (e), and (f) represent the 95 % confidence interval (using a standard deviation) from the daily mean, whereas the blue horizontal dashed line represents**





the yearly mean, the brown horizontal dashed line represents the yearly median, and the black horizontal dotted line represents a yearly standard deviation (see Table S2.1 and Fig. S2.2 present the Kulen meteorological station's annual and monthly meteorological data. Figs. S3.1–S3.3 shows monthly mean soil conditions by land-cover class, and Fig. S4.1 depicts correlations
between meteorological and soil conditions).

### 3.2 Species diversity

A total of 343 observations (292 trees and 51 seedlings) from 47 woody species (including 13 seedling species) and 32 families (including seven seedling families) were identified from the nine plots (Table S5.1). The average species richness ($S_R$) per plot for the EF, RF, and CP were 17, 13, and 4, respectively. The top five dominant species in EF accounted for 46 % of the

individuals: (*Mesua ferrea* (n = 18), *Diospyros bejaudii* (n = 12), *Litchi chinensis* (n = 11), *Vatica odorata* (n = 11), and *Hydnocarpus annamensis* (n = 8). In RF, the most dominant species were *Vatica odorata* (n = 54), *Nephelium hypoleucum* (n = 14), *Benkara fasciculata* (n = 12), *Garcinia oliveri* (n = 12), and *Mesua ferrea* (n = 5), comprising 61 % of individuals. Naturally, within the CP, the most abundant species was *Anacardium occidentale* (n = 46), the only species found when excluding seedlings. Among seedlings, except for *Anacardium occidentale,* we also found *Strychnos axillaris* (n = 3),

*Nephelium hypoleucum* (n = 1)*, Melodorum fruticosum* (n = 1)*, Maclura cochinchinensis* (n = 1)*,* and *Catunaregam tomentosa* (n = 1). Furthermore, fast-growth species, as described by Ha (2015) (*WD* < 0.6 g cm$^{-3}$), accounted for 40 % of EF and 44 % of RF of their total species composition.

The Shannon-Wiener index ranged from 0.31–2.68 across all plots, with the highest and lowest values observed in EF1 and CP2 (Table S5.2). EF showed the highest mean $S_H$ (2.48 ± 0.33), followed by RF (1.97 ± 0.45), whereas CP was dominated

by *Anacardium occidentale*, and thus it has a very low $S_H$ (0.61 ± 0.46).

**Table 2. Mean values and statistics of ecosystem characteristics in the different land-cover classes.**

| Group | Variables | Land cover | | | | | | | ANOVA | Tukey HSD | | |
|---|---|---|---|---|---|---|---|---|---|---|---|---|
| | | | | | | | | | | CP & EF | RF & EF | RF & CP |
| | | EF (Mean ± SD) | n | RF (Mean ± SD) | n | CP (Mean ± SD) | n | | p-value | p-value | p-value | p-value |
| Species diversity | $S_R$ (with seedling species, count per plot) | 17 ± 4 | 3 | 13 ± 2 | 3 | 4 ± 3 | 3 | | - | - | - | - |
| | $S_R$ (without seedling species, count per plot) | 13 ± 2 | 3 | 10 ± 3 | 3 | 1 ± 0 | 3 | | - | - | - | - |
| | $S_H$ (with seedling species, unitless) | 2.48 ± 0.33 | 3 | 1.97 ± 0.45 | 3 | 0.61 ± 0.46 | 3 | | - | - | - | - |





| | | | | | | | | | | |
|---|---|---|---|---|---|---|---|---|---|---|
| Leaf functional traits | $Chl_{cwm}$ (mg g$^{-1}$) | $9.14 \pm 3.45$ | 109 | $7.56 \pm 2.03$ | 137 | $4.99 \pm 0.66$ | 46 | * | * | 0.39 | 0.08 |
| | $LDMC_{cwm}$ (mg g$^{-1}$) | $398.43 \pm 72.24$ | 109 | $370.12 \pm 94.96$ | 137 | $407.64 \pm 21.68$ | 46 | 0.52 | 0.50 | 0.93 | 0.70 |
| | $SLA_{cwm}$ (m$^2$ kg$^{-1}$) | $18.18 \pm 2.86$ | 109 | $14.88 \pm 2.06$ | 137 | $11.99 \pm 1.45$ | 46 | ** | ** | * | 0.06 |
| Stand structures | $DBH$ (cm) | $18.0 \pm 20.1$ | 109 | $5.8 \pm 4.3$ | 137 | $13.0 \pm 3.9$ | 46 | *** | 0.14 | *** | *** |
| | $H$ (m) | $17.0 \pm 13.3$ | 109 | $7.4 \pm 3.8$ | 137 | $6.3 \pm 1.0$ | 46 | *** | *** | *** | 0.93 |
| | Maximum $H$ (m) | 52.0 | 109 | 18.6 | 137 | 7.8 | 46 | - | - | - | - |
| | Wood density (g cm$^{-3}$)$^†$ | $0.74 \pm 0.17$ | 109 | $0.72 \pm 0.15$ | 137 | $0.45 \pm 0.00$ | 46 | *** | *** | 0.56 | *** |
| | Stem density $DBH > 1$ cm (ha$^{-1}$)$^{††}$ | $6216 \pm 2177$ | 3 | $10859 \pm 4999$ | 3 | $1067 \pm 440$ | 3 | - | - | - | - |
| | Stem density $DBH > 5$ cm (ha$^{-1}$)$^{††}$ | $1016 \pm 533$ | 3 | $2193 \pm 895$ | 3 | $1067 \pm 440$ | 3 | - | - | - | - |
| | Stem density $DBH \geq 10$ cm (ha$^{-1}$)$^{††}$ | $550 \pm 505$ | 3 | $293 \pm 6$ | 3 | $600 \pm 164$ | 3 | - | - | - | - |
| | $BA$ (m$^2$ ha$^{-1}$) | $26 \pm 4$ | 3 | $17 \pm 5$ | 3 | $12 \pm 4$ | 3 | - | - | - | - |
| | $BA$ (m$^2$ ha$^{-1}$, $DBH \geq 5$ cm) | $24 \pm 4$ | 3 | $12 \pm 2$ | 3 | $12 \pm 3$ | 3 | - | - | - | - |
| | $BA$ (m$^2$ ha$^{-1}$, $DBH \geq 10$ cm) | $21 \pm 4$ | 3 | $4 \pm 1$ | 3 | $9 \pm 1$ | 3 | - | - | - | - |
| | $DWB$ (Total) (Mg ha$^{-1}$) | $27.5 \pm 12.4$ | 3 | $4.8 \pm 7.0$ | 3 | $0.4 \pm 0.2$ | 3 | - | - | - | - |
| | $AGB_f$ (Mg ha$^{-1}$) | $239 \pm 92$ | 3 | $42 \pm 10$ | 3 | $71 \pm 22$ | 3 | - | - | - | - |
| | $AGB_{wd}$ (Mg ha$^{-1}$) | $336 \pm 168$ | 3 | $78 \pm 25$ | 3 | $182 \pm 57$ | 3 | - | - | - | - |
| | $AGB_h$ (Mg ha$^{-1}$) | $312 \pm 184$ | 3 | $54 \pm 14$ | 3 | $17 \pm 5$ | 3 | - | - | - | - |
| | $LAI_C$ (m$^2$ m$^{-2}$) | $4.62 \pm 0.50$ | 21 | $4.66 \pm 0.70$ | 21 | $2.52 \pm 0.42$ | 21 | *** | *** | 1.00 | *** |
| | $LAI_T$ (m$^2$ m$^{-2}$) | $6.16 \pm 0.67$ | 21 | $5.57 \pm 0.76$ | 21 | $3.07 \pm 0.61$ | 21 | *** | *** | 0.08 | *** |
| | Annual mean $fPAR^‡$ | $0.97 \pm 0.01$ | 364 | $0.96 \pm 0.01$ | 365 | $0.76 \pm 0.06$ | 359 | *** | *** | * | *** |
| Soil conditions | Annual mean $SWC^‡$ (m$^3$ m$^{-3}$) | $0.23 \pm 0.06$ | 364 | $0.14 \pm 0.03$ | 365 | $0.21 \pm 0.05$ | 363 | *** | *** | *** | *** |
| | Annual mean $Ts^‡$ (°C) | $24.3 \pm 1.2$ | 364 | $24.2 \pm 1.3$ | 365 | $25.8 \pm 1.5$ | 363 | *** | *** | *** | *** |
| | Annual mean $ECs^‡$ (dS m$^{-1}$) | $0.039 \pm 0.015$ | 268 | $0.032 \pm 0.013$ | 40 | $0.025 \pm 0.003$ | 260 | *** | *** | *** | *** |

**Note: Abbreviations used in the table: EF = evergreen forests, RF = regrowth forests, CP = cashew plantations, $S_R$ = species richness (only woody seedling species), $S_H$ = Shannon-Wiener index, $Chl_{cwm}$ = community-weighted mean of chlorophyll a and b content, $LDMC_{cwm}$ = community-weighted mean of leaf dry matter content, $SLA_{cwm}$ = community-weighted mean of specific leaf area, $DBH$ = tree's diameter at breast height, $H$ = tree height, $BA$ = stand basal area, $AGB_f$ = aboveground biomass computed by adopted functions, $AGB_h$ = aboveground biomass computed by $H$ and $DBH$ power-law relationship, $AGB_{wd}$ = aboveground biomass based on equations Eqs. (9–11) with species-specific wood density updated for our woody tree species, $LAI_C$ = canopy leaf area index, $LAI_T$ = total leaf area index, $fPAR$ = fraction of photosynthetically active radiation, $SWC$ = soil water content, $Ts$ = soil temperature, $ECs$ = soil saturation extract electrical conductivity, SD = a standard deviation, ANOVA = one-way analysis of variance, Tukey HSD = Tukey's Honestly Significant Difference test. Statistically significant code for ANOVA and Tukey HSD test: '***' p-value < 0.001, '**' p-value < 0.01, '*' p-value < 0.05, and "-" not available. $^†$The species-specific wood density was derived from the ICRAF Database (2022) and Zanne et al. (2009). $^{††}$Extrapolated values for one hectare were obtained from sampling $DBH$ class subplots. $^‡$Daily mean values were used to calculate the reported variables.**





### 3.3 Leaf functional traits and diversity

The mean specific leaf area for all 30 species was $16.97 \pm 5.30$ m$^2$ kg$^{-1}$, with *Hydnocarpus annamensis* having the highest *SLA* ($36.67 \pm 5.20$ m$^2$ kg$^{-1}$) and *Capparis micracantha* the lowest ($10.46 \pm 3.28$ m$^2$ kg$^{-1}$). For *Chl*, the mean value was $10.28 \pm 4.17$ mg g$^{-1}$, with *Hydnocarpus annamensis* having the highest value ($25.75 \pm 5.28$ mg g$^{-1}$) and *Anacardium occidentale* the lowest ($4.86 \pm 4.93$ mg g$^{-1}$). Finally, for *LDMC* the mean value was $378.96 \pm 143.26$ mg g$^{-1}$, with *Mesua ferrea* and *Hydnocarpus annamensis* having the highest ($486.90 \pm 25.03$ mg g$^{-1}$) and lowest ($139.92 \pm 20.19$ mg g$^{-1}$) values, respectively. For detailed descriptions of leaf functional traits of all species and plots, please refer to Tables S6.1–S6.3.

There were statistical differences in mean $SLA_{cwm}$ (p-value < 0.002) and $Chl_{cwm}$ (p-value < 0.018) among the three land-cover classes, whereas there was no significant difference in the mean $LDMC_{cwm}$ (p-value = 0.52) (Table 2). $SLA_{cwm}$ and $Chl_{cwm}$ were highest in EF ($18.18 \pm 2.86$ m$^2$ kg$^{-1}$, and $9.14 \pm 3.45$ mg g$^{-1}$) followed by RF ($14.88 \pm 2.06$ m$^2$ kg$^{-1}$, and $7.56 \pm 2.03$ mg g$^{-1}$) and CP ($11.99 \pm 1.45$ m$^2$ kg$^{-1}$, and $4.99 \pm 0.66$ mg g$^{-1}$). However, for $LDMC_{cwm}$ the highest value was observed in CP, with a value of $407.64 \pm 21.68$ mg g$^{-1}$ ($398.43 \pm 72.24$ mg g$^{-1}$ for EF, $370.12 \pm 94.96$ mg g$^{-1}$ for RF). See Table S6.4 for data sources and shared percentages of species trait values used to compute $SLA_{cwm}$, $Chl_{cwm}$, and $LDMC_{cwm}$.

### 3.4 Stand structure attributes

#### 3.4.1 *DBH-H* relationship

The 292 sampled woody trees in the nine inventory plots had a mean *DBH* of $11.5 \pm 13.9$ cm and a mean *H* of $10.8 \pm 9.8$ m (Fig. S7.1). The maximum *H* of 52.0 m and the maximum *DBH* of 102.3 cm were both observed in EF. RF and CP had maximum *H* of 18.6 m and 7.8 m, and maximum *DBH* of 23.1 cm and 18.8 cm respectively. Comparing land-cover classes, EF had both the highest mean and the highest variability in *DBH* ($18.0 \pm 20.1$ cm) and highest *H* ($17.0 \pm 13.3$ m), while CP had a mean *DBH* of $13.0 \pm 3.9$ cm, which was double that of RF ($5.8 \pm 4.3$ cm). CP had slightly higher mean *H* values than RF, whereas RF had higher variability (RF at $7.4 \pm 3.8$ m; CP at $6.3 \pm 1.0$ m). In addition, the ANOVA confirmed statistically significant differences in mean *DBH* and mean *H* among the three land-cover classes (Table 2). The Tukey HSD test further revealed differences in mean *DBH* for RF & EF and RF & CP (p-value < 0.001), as well as in mean *H* for CP & EF and RF & EF (p-value < 0.001). In contrast, the test showed no statistically significant differences between the mean *DBH* for CP & EF (p-value = 0.14) and mean *H* for CP & RF (p-value = 0.93) (Table 2).

Strong positive relationships between *DBH* and *H* were observed in both EF and RF. For EF, 92 % of the variation in *H* can be explained by the variation in *DBH*, whereas for RF and CP, it was 78 % and 51 %, respectively (Table S7.1). The power-law relationships between *DBH* and *H* further indicated that the $K_1$ and $K_2$ values for EF and RF were similar, whereas the values for CP were much lower (Fig. 3). For a plot-level analysis of relationships between ln(*DBH*) and ln(*H*), see Fig. S7.2 and Table S7.2.



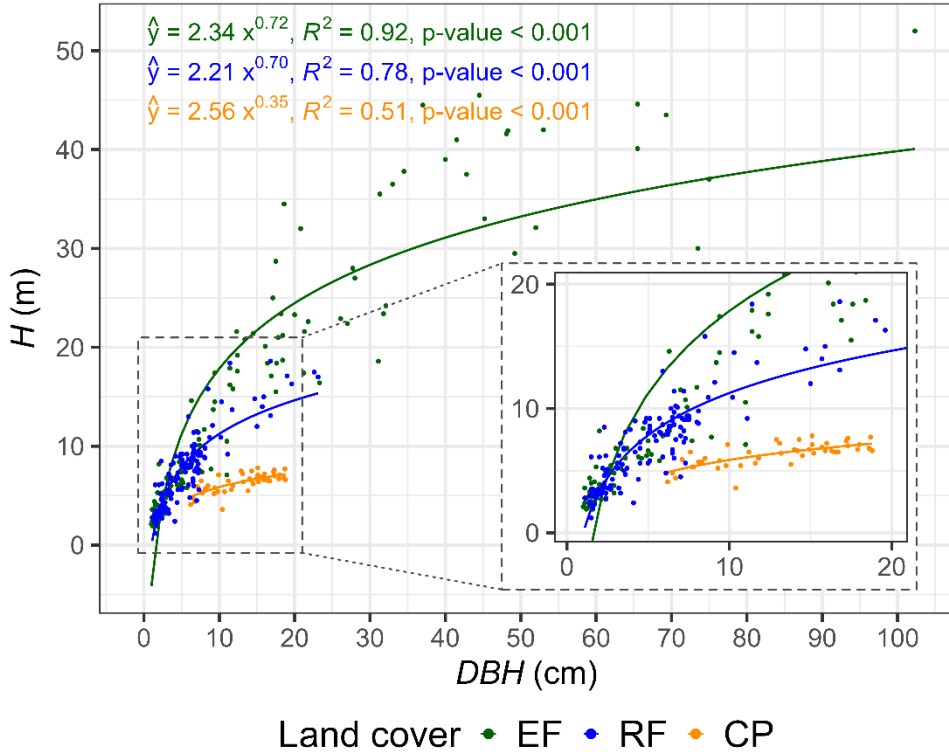

310

**Figure 3. Relationship between diameter at breast height (*DBH*) (cm) and tree height (*H*) (m) for evergreen forests (EF), regrowth forests (RF), and cashew plantations (CP) in Kulen. Figure shows the derived power-law intercept ($K_1$) and slope ($K_2$) values for EF, RF, and CP.**

### 3.4.2 Aboveground and deadwood biomass, stem density, and basal area

315    The $AGB_{wd}$ method consistently yielded higher aboveground biomass across all land-cover classes compared to the $AGB_f$ and $AGB_h$ methods. Significant variations in estimated $AGB$ values were observed among land-cover classes for all methods; EF had values in the range of 239–336 Mg ha$^{-1}$, and RF had values in the range of 42–78 Mg ha$^{-1}$ (Table 2, Fig. 4). The main difference between the methods was that $AGB_h$, based on local scale $K_1$ and $K_2$ values (Table S7.2, Fig. S7.2), had substantially lower $AGB$ for CP compared to the other two methods $(AGB_{wd}$: 182 ± 57 Mg ha$^{-1}$, $AGB_f$: 71 ± 22 Mg ha$^{-1}$, $AGB_h$: 17 ± 5 Mg

320    ha$^{-1}$, see Fig. S7.3). Further plot-level results are available in Figs. S7.4–S7.5. Additionally, the mean total $DWB$ was 27.5 ± 12.4 Mg ha$^{-1}$ in EF, 4.8 ± 7.0 Mg ha$^{-1}$ in RF, and 0.4 ± 0.2 Mg ha$^{-1}$ in CP. See Table A1 for the contribution of lying and standing $DWB$ to total $DWB$.





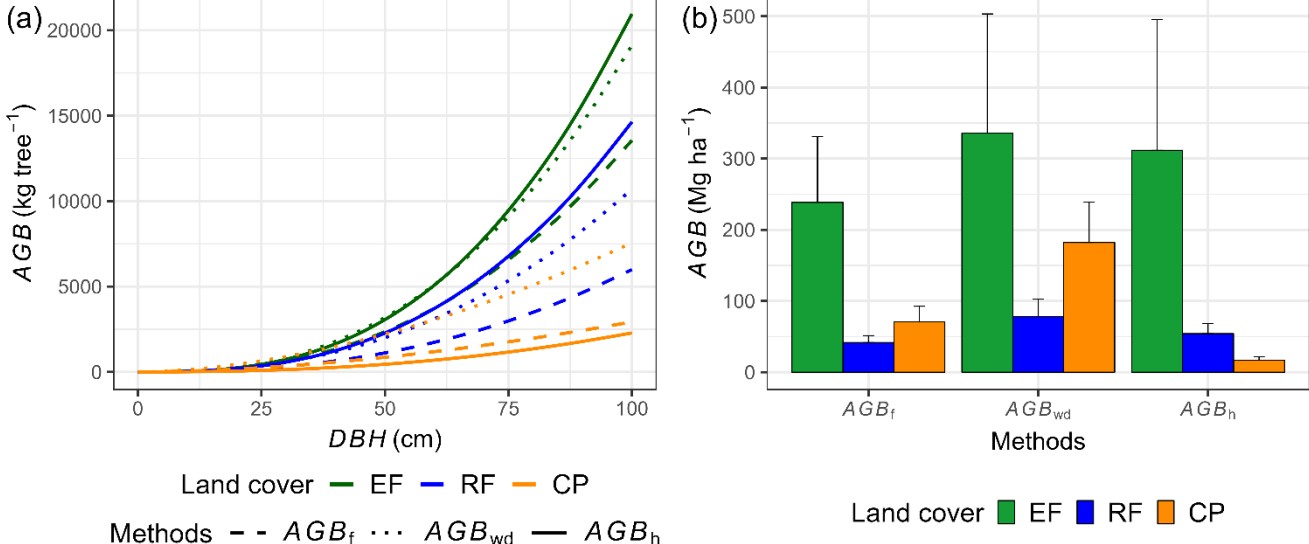

**Figure 4. Power-law relationships between aboveground biomass (*AGB*) of *AGB*$_f$, *AGB*$_h$, and *AGB*$_{wd}$ and diameter at breast height (*DBH*) for each land-cover class (a), along with the corresponding results of *AGB* estimation (b). *AGB*$_f$ represents aboveground biomass estimated by adopted functions, *AGB*$_{wd}$ represents aboveground biomass estimated by adopted functions utilizing species-specific wood density, and *AGB*$_h$ represents aboveground biomass estimated by the *DBH* and tree height (*H*) relationship, in conjunction with species-specific wood density, for the study site. The error bar in (b) represents a standard deviation.**

The stem density per hectare (*DBH* > 5 cm) was twice as high in the RF compared to EF and CP (Fig. 5). This higher stem density per ha was primarily attributed to the *DBH* class of 5–15 cm. RF had a significantly larger *BA* ($17 \pm 5$ m$^2$ ha$^{-1}$) than the CP ($12 \pm 4$ m$^2$ ha$^{-1}$), despite having a smaller mean *DBH* (Table 2). Interestingly, only 5 % of the stems with a *DBH* > 30 cm contributed to approximately 75 % of the total *AGB*$_h$, 234 Mg ha$^{-1}$ out of 312 Mg ha$^{-1}$. The main *DBH* class contributing to the *AGB* in RF and CP was 5–15 cm, accounting for 62 % and 71 % of the total *AGB* in RF and in CP, respectively. Refer to Supplementary Table S7.3 for shared stem density percentages per hectare across *DBH* classes, and Table S7.4 for shared percentages of *AGB*$_h$ categorized by *DBH* class.



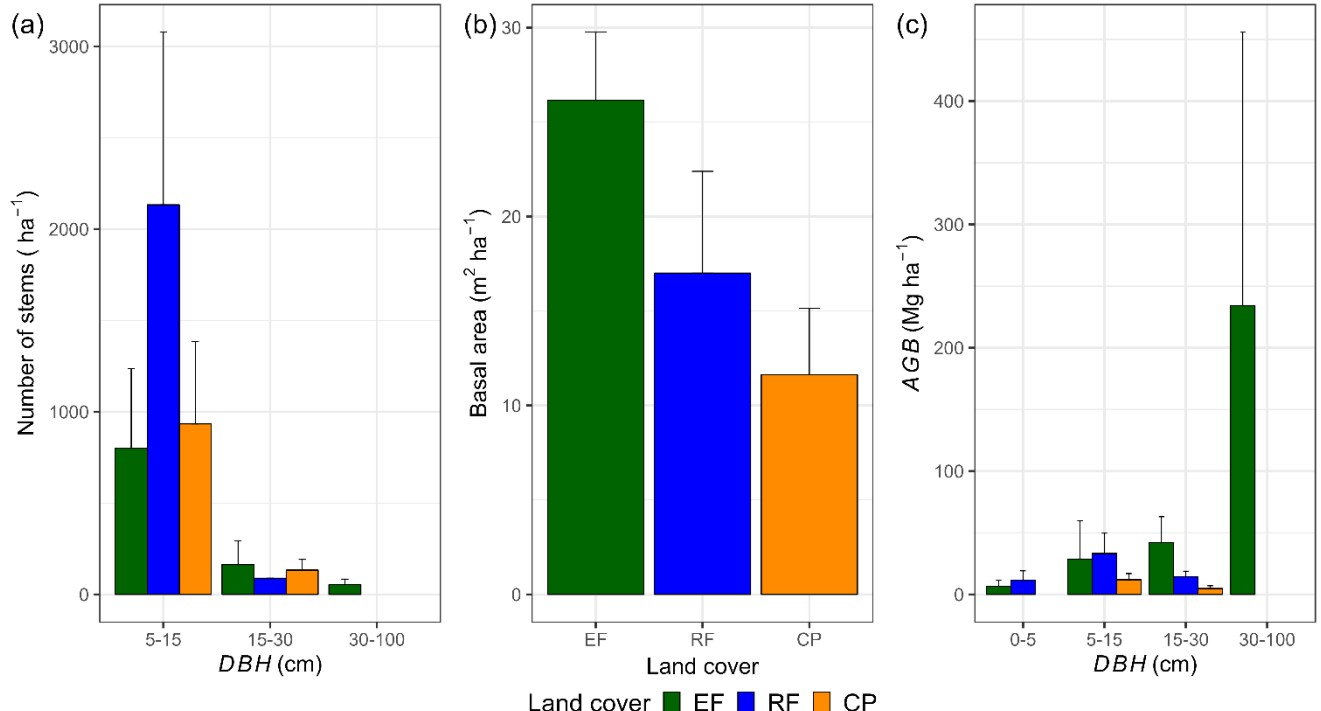

**Figure 5. Estimations per land-cover class of a mean number of stems per hectare (a), basal area ($BA$, m² ha⁻¹) (b), and mean aboveground biomass separated by the different diameters at breast height ($DBH$) classes (c). In (c), the contribution of different $DBH$ classes to the mean aboveground biomass estimated by the $AGB_h$ method was used in this calculation. The error bars in the figure represent one standard deviation.**

### 3.5 *LAI* and *fPAR*

The mean total leaf area index values were $6.16 \pm 0.67$ m² m⁻² for EF, $5.57 \pm 0.76$ m² m⁻² for RF, and $3.07 \pm 0.61$ m² m⁻² for CP. The mean canopy *LAI* values were $4.62 \pm 0.5$ m² m⁻² for EF, $4.66 \pm 0.70$ m² m⁻² for RF, and $2.52 \pm 0.42$ m² m⁻² for CP. The ANOVA analysis revealed a significant difference in mean $LAI_T$ and mean $LAI_C$ among the three land-cover classes, while the Tukey HSD test did not find a significant difference in mean $LAI_T$ and mean $LAI_C$ between EF and RF (Table 2). The phenology of both $LAI_T$ and $LAI_C$ revealed a similar pattern in EF and RF, with peak and base values in June and March, respectively (Fig. 6a–b, Table S8.1 in Supplementary Subsection 8 for their descriptive statistics). The $LAI_T$ and $LAI_C$ patterns for CP resembled those of EF and RF but also had a strong decrease in April. Furthermore, the understory *LAI* ($LAI_U$; the difference between $LAI_T$ and $LAI_C$) for the various land-cover classes indicates that the ground vegetation highly contributes to $LAI_T$ for EF and RF, while the contribution was minor for CP (Fig. 6c). In particular, the $LAI_U$ mean values within a year were approximately $1.54 \pm 0.57$ m² m⁻² for EF (25 %), $0.91 \pm 0.36$ m² m⁻² for RF (16 %), and $0.55 \pm 0.39$ m² m⁻² for CP (18 %). A general trend of high contribution $LAI_U$ to $LAI_T$ in June and low contribution in April was apparent for all land-cover classes.



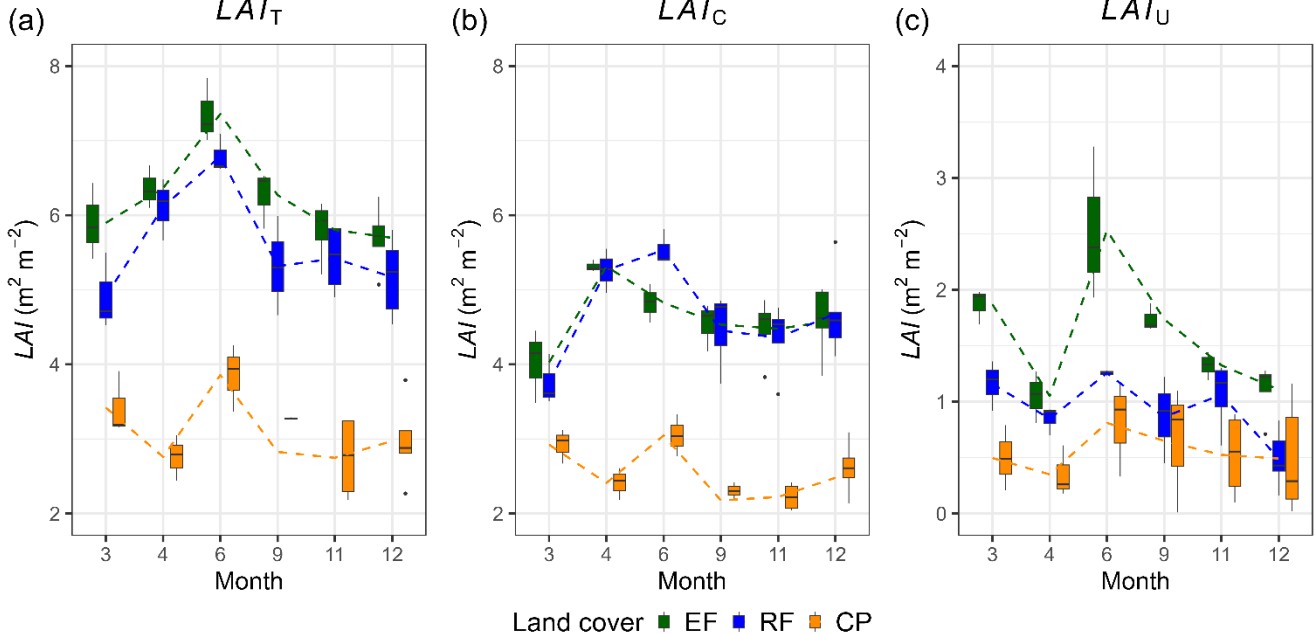

355

**Figure 6. Total leaf area index ($LAI_T$, m² m⁻²), canopy leaf area index ($LAI_C$, m² m⁻²), and understory leaf area index ($LAI_U$, m² m⁻²), their variations across different months within a year for evergreen forests (EF), regrowth forests (RF), and cashew plantations (CP). The lines in the graph represent the connection between the mean *LAI* values from one month to another.**

The observed mean annual *fPAR* for EF, RF, and CP was high: $0.97 \pm 0.01$, $0.96 \pm 0.01$, and $0.76 \pm 0.06$, respectively (Table

360 2). The values of EF and RF exhibited minimal fluctuations throughout the year, whereas the *fPAR* of CP ranged between 0.55 and 0.93 (Fig. 7). Like *LAI,* the annual mean *fPAR* among EF, RF, and CP were statistically significantly different according to both the ANOVA test and Tukey HSD's tests.



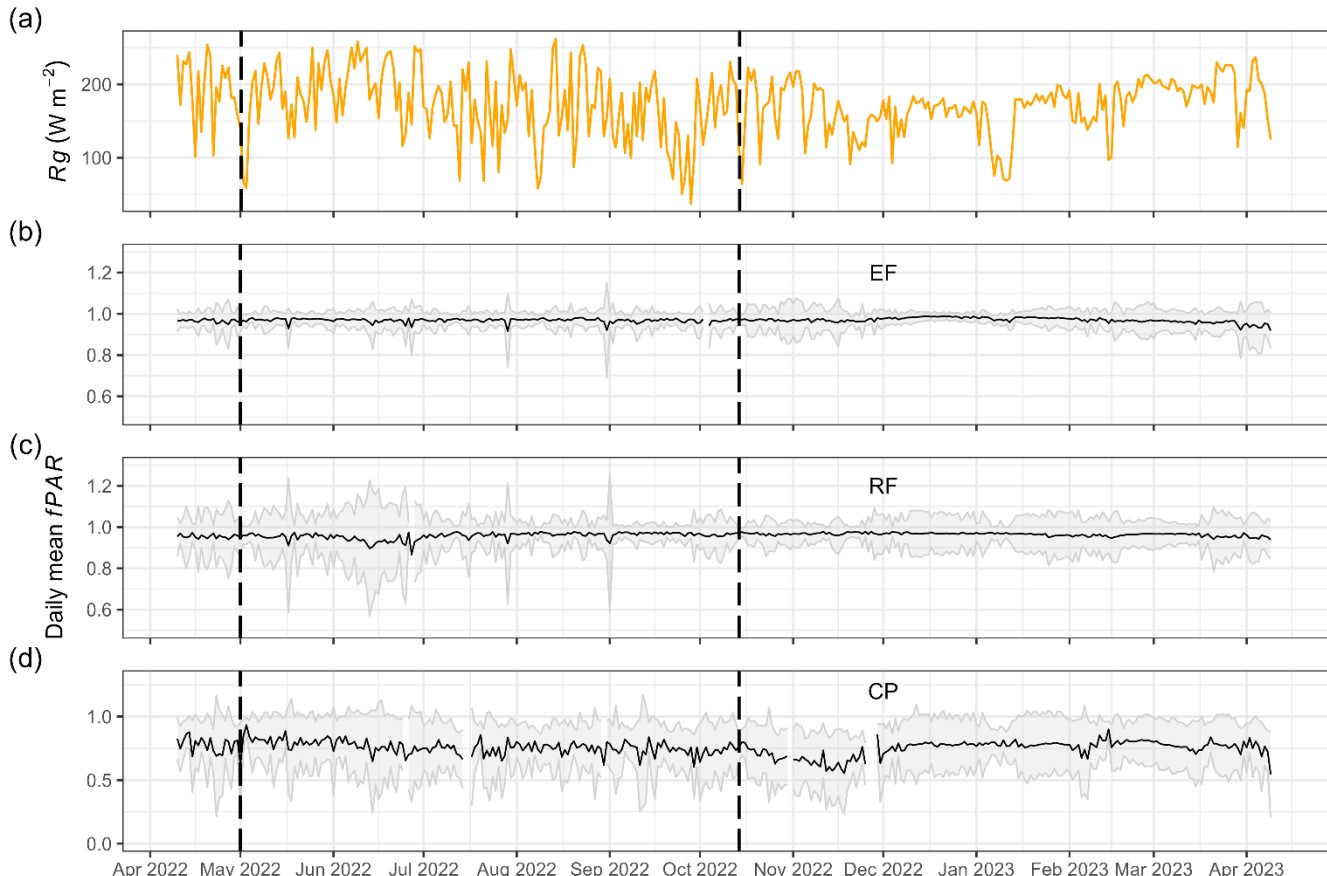

**Figure 7. Daily mean global radiation ($Rg$, W m$^{-2}$) (a) and daily mean $fPAR$ for evergreen forests (EF) (b), regrowth forests (RF) (c), cashew plantations (CP) (d) from April 11, 2022, to April 9, 2023 at Kulen. The shaded area represents one standard deviation from the mean, computed using the ten $PAR$ sensors installed in each land-cover class.**

### 3.6 $AGB_h$ relationships with $LAI_T$, $SLA_{cwm}$, and $S_R$

We observed positive relationships between aboveground biomass and three pivotal ecosystem characteristics: $LAI_T$, $S_R$, and $SLA_{cwm}$ determining 76 %, 72 %, and 68 % of the variability in $AGB$, respectively (Fig. 8, Table S9.1 for statistical regression tables). $LAI_T$ exhibited strong positive correlations with $SLA_{cwm}$, $S_R$, and $AGB$, with the Pearson correlation coefficient in the range of 0.67–0.85. $SLA_{cwm}$ had a positive correlation with $S_R$ and $AGB$. Furthermore, additional insights regarding the Pearson correlation matrix depicting relationships among various ecosystem characteristics are presented in Fig. S9.1.





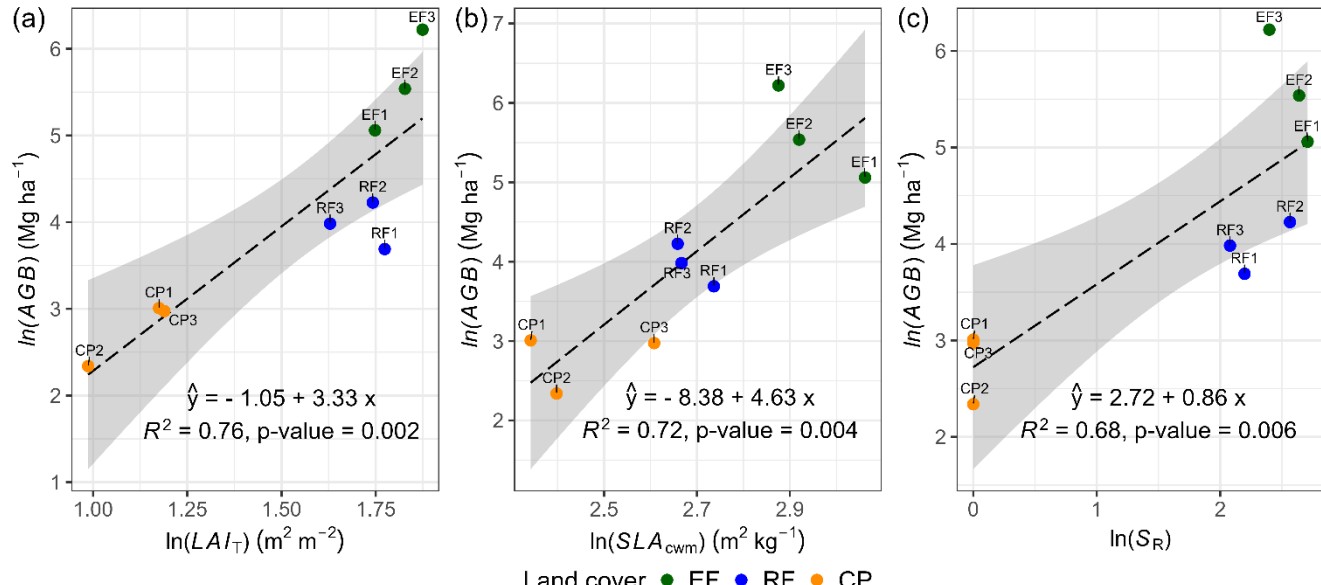

**Figure 8. Ordinary least squares regression showing the effect of mean total *LAI* (*LAI*$_T$, m$^2$ m$^{-2}$), mean *SLA*$_{cwm}$ (m$^2$ kg$^{-1}$), and species richness (*S*$_R$, count per plot) on *AGB*. Mean *LAI*$_T$ is a mean ground *LAI* measurement, *S*$_R$ is a woody species count excluding seedlings in a plot, and *AGB* is *AGB*$_h$ whose estimation was based on the *DBH-H* relationship.**

## 4 Discussions

### 4.1 Importance of tropical field data

Numerous studies have emphasised the essential role of field-observed data for empirically elucidating the complexities of tropical forest ecosystems (Fischer et al., 2016; Clark et al., 2017). These data are crucial for understanding how land-use and land-cover changes affect forest ecosystems, for assessing biodiversity, for mapping, and for quantification of ecosystem services, and for enhancing remote sensing and ecosystem modelling techniques. In our case, the observed dataset also allows for pairwise comparisons of ecosystem characteristics against the pristine conditions of the tropical evergreen forests, a perspective often missing in studies focused on a single land-cover class. Given the critical issue of land conversion in Southeast Asia, where natural forests are frequently transformed into agricultural lands, our data are pivotal for studying the ecological shifts of such land cover changes. Despite not being similar to the nearby lowlands (Chim et al., 2021), the meteorological conditions at the field site are characteristic of the tropical monsoon climate of Southeast Asia (Thoeun, 2015), indicating that the conclusions drawn from field site data may represent the larger region.

### 4.2 Soil conditions

The difference in soil temperature between the forested land-cover classes (EF and RF) and the cashew plantations (Table 2) aligns with prior studies by van Haren et al. (2013) and Geng et al. (2022) and can be explained by the substantial difference





in interception of incoming radiation between these ecosystems (Fig. 6). The multi-layered canopies and the dense layer of deadwood and litterfall, effectively prevent direct sunlight from reaching the ground. This natural shield reduces the impact of

global radiation, thereby maintaining cooler soil surface temperatures (Senior et al., 2018). Conversely, CP has a simpler canopy structure, predominantly featuring a single layer of cashew trees of similar age. The understory in these areas is sparser, and the reduction in deadwood, due to management, facilitates greater global radiation penetration and elevates soil temperatures.

Our observed annual mean soil water content across the three land-cover classes (0.14–0.23 m$^3$ m$^{-3}$) is consistent with earlier

findings (Rodell et al., 2004; Wang et al., 2012; Horel et al., 2022). Variations in *SWC* among these classes may stem from differences in their stand structural complexity (vegetation cover and root system) and soil properties (organic matter content and texture) (Pickering et al., 2021; Tang et al., 2021). The higher *SWC* in evergreen forests compared to regrowth forests is attributed to their dense and multilayered vegetation cover, which reduces global radiation and temperature at the forest floor, thereby reducing evaporation and maintaining topsoil moisture (Fig. 6). In addition, the complex root systems of primary

forests enhance water retention by creating channels and pores in the soil, while organic matter from deadwood and litterfall further enhances soil water retention, particularly during arid conditions (Luo et al., 2023). Another explanation could be the soil texture, as our field investigation observed that cashew plantations are all on sandier soils with lower water-holding capacity, leading to decreased *SWC* (Ibrahim and Alghamdi, 2021). Nevertheless, further examination of soil samples is necessary to accurately measure the specific soil properties in each land-cover class.

The analysis of soil electrical conductivity categorized the soils as non-saline across the land-cover classes. evergreen forests had higher *ECs* than cashew and regrowth forests, potentially indicating larger nutrient availability (Omuto et al., 2020). This higher nutrient availability in evergreen forests may be linked to greater organic matter decomposition, species richness, higher soil moisture content, and no history of being clear-cut, which could lead to nutrient losses via run-off during the phase without vegetation (Austin et al., 2004; Vestin et al., 2020; Guo et al., 2023b).

**4.3 Species diversity**

The mean species richness and biodiversity (the Shannon-Wiener index) of evergreen forests and regrowth forests (Table 2) were similar to several previous studies of evergreen forests (Zin and Mitlöhner, 2020; Theilade et al., 2022; Tynsong et al., 2022). However, species richness was lower compared to the most diverse rainforests in South America and Southeast Asia, where often > 250 species ha$^{-1}$ have been reported (Mohd Nazip, 2012; ter Steege et al., 2023) and the Shannon-Wiener index

was lower than for some moist evergreen and humid lowland forests in Southeast Asia (Mohd Nazip, 2012; Zin and Mitlöhner, 2020). These tropical rainforests may have more species because of their larger forest patch sizes and higher rainfall, compared to the relatively isolated monsoon forest at the top of Kulen, surrounded by agricultural areas (Galanes and Thomlinson, 2009). The relatively low $S_H$ may also be explained by the high proportion of the top five dominant species in each land cover, accounting for over 50 % of total stems in their communities. Another possible reason could be the limited number of sample

plots, which may not fully capture the overall species composition and distribution in these forests. Tropical tree species





composition is markedly influenced by biogeography and disturbance history, showing significant local variations even over short distances (Whitmore, 1998; Van and Cochard, 2017). This emphasizes the necessity for comprehensive field data sampling to accurately assess the species richness and evenness of these highly diverse plant communities. The comparison between $S_R$ and $S_H$ of EF and RF with previous studies is presented in Table S10.1– S10.2.

### 430   4.4 Leaf functional traits and diversity

Specific leaf area, leaf dry matter content, and chlorophyll content are all key leaf traits in the leaf economic spectrum, and carry diverse implications for understanding carbon sequestration, resource availability, successional stages, and environmental responses (Wright et al., 2004; Gao et al., 2022). The $SLA_{cwm}$ of EF exceeded the mean values of tropical forests in Bolivia, Brazil, Costa Rica, and China (Finegan et al., 2015; Wang et al., 2016). The $SLA_{cwm}$ of RF was somewhat higher

than the mean of neotropical regrowth forests, but still within the range (Poorter, 2021). $SLA_{cwm}$ in CP was greater than the range value in Parakou, Benin, but fell within the range reported for 15 cashew varieties in Karnataka, India (Akossou et al., 2016; Mog and Nayak, 2018). Furthermore, our observations emphasize the significant consequences of transitioning from EF to RF or CP, resulting in a substantial reduction in actual values and diversity in $SLA_{cwm}$, reflecting a reduction in both ecosystem productivity and resilience to disturbances (Liu et al., 2023). The higher $SLA_{cwm}$ in EF, suggests higher

photosynthetic capacity, especially in shaded environments, due to its dense canopy cover and abundant resource availability (water and nutrients) for plant growth (Green et al., 2020). High $SLA_{cwm}$ values also link to faster turnover and promote nutrient cycles, carbon sequestration, and nutrient use efficiency in forest ecosystems (Guerrieri et al., 2021). The lower $SLA_{cwm}$ values of RF and CP may be attributed to limited water and nutrient availability in the soil because of high competition in those ecosystems. The notable reduction in $SLA_{cwm}$ caused by the shift from EF to RF or CP underlines the profound impact land-

cover change has on both $SLA_{cwm}$ and, consequently, ecosystem productivity, resilience, and functioning, highlighting the impact of land-cover change on ecosystem function.

Leaf dry matter content is a measure of construction cost per fresh weight mass unit, and it serves as a metric for a plant's resource use strategy and resilience to environmental stresses (Guo et al., 2023a). The higher $LDMC_{cwm}$ in EF compared to that of RF indicates a conservative resource usage, longer leaf lifespan, and increased carbon sequestration, implying higher

ecosystem stability and function for EF (Rawat et al., 2021). Conversely, the highest $LDMC_{cwm}$ in CP, is attributed to cashew monoculture and the species' high resilience to environmental stress, especially in nutrient-poor soils and water-stressed conditions (Bezerra et al., 2007). This study emphasises EF's increased stress tolerance, conservative resource utilisation and greater carbon sequestration compared to RF, while also emphasizing cashew as a highly proficient species in environmental stress tolerance.

Chlorophyll is essential for photosynthesis and serves as a crucial indicator of a plant's photosynthetic capacity, profoundly influencing overall growth (Stirbet et al., 2020). In this study, $Chl_{cwm}$ in EF and RF falls within the range observed in Chinese forest ecosystems but surpasses the mean $Chl_{cwm}$ in those ecosystems (Li et al., 2018). Our CP had lower $Chl_{cwm}$ than EF and RF due to less light competition and higher temperatures, which could lead to photoinhibition and lowered leaf chlorophyll



content. The elevated $Chl_{cwm}$ seen in EF can be attributed to the well-developed and dense canopy structure, which creates a
light-shaded environment. This prompts plants to invest more in chlorophyll production, enhancing light harvesting efficiency.
Meanwhile, RF, experiencing intense competition for light in early successional stages, may exhibit lower chlorophyll levels
as resources prioritize vertical growth over chlorophyll production (Laurans et al., 2014).

### 4.5 Stand structure attributes

### 4.5.1 Tree height and diameter at breast height

Our mean *DBH* of evergreen forests is comparable to mature tropical forests in Vietnam and falls within the pantropical range,
while regrowth forests have a slightly higher mean *DBH* than tropical secondary forests in Sarawak, Malaysia (Brown, 1997;
Kenzo et al., 2009; Yen and Cochard, 2017). In contrast, cashew plantations show a significantly lower mean *DBH* compared
to older counterparts in Kampong Cham, Cambodia (Avtar et al., 2013). Moreover, observed species in our evergreen forests,
such as *Dipterocarpus costatus*, *Sandoricum indicum*, *Mesua ferrea*, *Nageia wallichiana*, and *Litchi chinensis* reach heights
of 40–52 m, similar to those found in Cambodia's central evergreen forests (Theilade et al., 2022). The loss of large-diameter
and tall trees in regrowth forests and cashew plantations, resulting from land use changes, substantially threatens critical
ecosystem functions, jeopardizing carbon storage, nutrient cycles, and biodiversity within these transformed landscapes (Díaz
et al., 2007; Lutz et al., 2018; Thiel et al., 2021).

### 4.5.2 Stem density and basal area

Our mean stem density per hectare of evergreen forests is consistent with previous studies in Cambodia, Vietnam, and in
Borneo, while regrowth forests show lower densities compared to those in the Yucatan Peninsula, Mexico (Slik et al., 2010;
Con et al., 2013; Román-Dañobeytia et al., 2014; Chheng et al., 2016; Theilade et al., 2022). Additionally, our stem density in
cashew plantations is similar to that of Isuochi, Nigeria, but significantly greater than that of Casamance, Senegal, due to their
differences in planting distance and management practices (Nzegbule et al., 2013; Ndiaye et al., 2020). The variation in stem
density between regrowth forests and evergreen forests reflects distinctive stages of succession. In the early succession stage
following clearance, open niches and resource abundance create a favourable environment for fast-growing and highly
reproductive early-succession species, resulting in higher stem density and heightened interspecies competition (Zhang et al.,
2020). As the forest matures, stem density naturally decreases as larger trees occupy more space, light, water, and nutrient
resources. This competition ultimately leads to the mortality of smaller trees, aligning with the power-law relationship between
stem density and biomass commonly observed in mature forests (Mrad et al., 2020). This natural process also alters species
composition, stand structure, habitat heterogeneity, and biomass of forests (Forrester et al., 2021). In cashew plantations, stem
density is controlled by humans to enhance cashew yield. This alteration in stand structure complexity influences interspecies
competition. These modifications also affect stand structure and interspecies competition, ultimately influencing the
biodiversity and functioning of the ecosystem.





Basal area, by incorporating both cross-sectional area and stem density in a given area, offers crucial insights into stand structure dynamics. The basal area of evergreen forests in our study aligns with those in northeast Cambodia and Pahang National Park, Malaysia, but falls below values reported for Laos, Cambodia's central plains and Vietnam's lowlands (Rundel, 1999; Sovu et al., 2009; Mohd Nazip, 2012; Chheng et al., 2016; Theilade et al., 2022). Regrowth forests have lower *BA* than evergreen forests, indicating early succession and disturbance (Ziegler, 2000). Still, our regrowth forest's *BA* exceeds that of

regrowth forest in Laos, while cashew plantations surpass plantations in Tanzania's (Sovu et al., 2009; Malimbwi et al., 2016). Basal area decreases significantly when EF is replaced with CP or RF, impacting biomass, productivity, stand structure, and structural complexity (Gea-Izquierdo and Sánchez-González, 2022). While tropical forests possess natural regenerative capabilities, RF may require several decades to achieve *BA* levels comparable to EF, highlighting the critical importance of conserving EF to maintain their ecological integrity and ecosystem services.

**4.5.3 *DBH-H* relationships and estimations of aboveground biomass**

The *DBH-H* relationship is crucial for understanding variations in tree growth rates, successional stage, aboveground biomass, and forest health (Kramer et al., 2023). Finding a strong positive *DBH-H* relationship may indicate disturbances within the ecosystem, as these by initiating gaps in the canopy provide opportunities for fast-growing species to establish and utilize increased light availability and resources within the ecosystem (Senf et al., 2020). Hence, the observed relationships between

EF and RF suggest a composition of fast-growing species and indicate that EF may have experienced past disturbances. Indeed, a windthrow in EF1 is reflected in its lowest $LAI_C$ among EF plots and a smaller mean *DBH* (Fig. S7.1a).

The lower *DBH-H* relationship in cashew plantations results from the growth strategy of the single species and management practices. In monocultures with uniformly aged cashew plants, competition for light and resources is comparable, resulting in a consistent resource distribution. Cashew's natural growth characteristics, with the species reaching up to 15 m in height and

a *DBH* of 100 cm under favourable conditions (Avtar et al., 2014), indicate a preference for investing resources in branches and stems over height, especially in low-light competition environments. However, our observations indicate significant variation in the *DBH-H* relationship among CP plots (low $R^2$ value in Fig. 3, Fig. S7.5g–i) which may have been influenced by their different management practices, such as spacing, pruning, and thinning. These practices impact the *DBH-H* relationship by minimizing light competition, resulting in a higher *DBH-H* ratio which also affects the relationship (Deng et

al., 2019; Bhandari et al., 2021).

Recent studies have emphasized the significant uncertainty in estimating plot-level aboveground biomass when directly applying a generic *AGB* allometric equation ($AGB_f$) due to variations in species composition and stand structure between the study site and the equation's origin (Feldpausch et al., 2011; Burt et al., 2020). To address this challenge, our study proposes an allometric approach ($AGB_h$) using local species-specific wood density and the *DBH-H* relationship at the study site. This

approach captures the unique characteristics of the site's species composition and stand structure (Ketterings et al., 2001; Nyirambangutse et al., 2017). Our locally adopted $AGB_h$ method produced estimates ~ 30 % higher than the generic $AGB_f$ for both EF and RF (Table 2, Fig. 4b). This is likely due to the combined effects of higher mean wood density and a stronger *DBH*





relationship, resulting in a more pronounced exponential growth response in *AGB* (Fig. 4a). Still, these ~ 30 % higher values align with the range reported in previous studies (Tables A2–A3). In contrast, in the CP case, our $AGB_h$ method produced

estimates less than a quarter of the generic $AGB_f$ method. The reason is that the $AGB_h$ method is less reliable when a weak *DBH-H* relationship is detected because it fails to accurately capture the overall tree size and volume. This is also reflected in the substantially larger uncertainty in the CP $AGB_h$ method as indicated by the standardized errors of the parameters within the *DBH-H* relationship (Table A4; Table S7.1). However, to fully validate the *AGB* allometric equations destructive field-observed data would be necessary. Therefore, future research should include direct field measurements of *AGB* to more

accurately validate the methods for these land-cover classes.

### 4.5.4 Deadwood biomass

Deadwood biomass indicates biodiversity and ecosystem health, supporting various species and ecosystem processes like carbon and nitrogen cycling, soil fertility enhancement, pollination, and erosion control (Parisi et al., 2018a; Santopuoli et al., 2021; Tláskal et al., 2021). In evergreen and regrowth forests, we found total deadwood biomass comparable to Cambodia and

Malaysia (Saner et al., 2012; Kiyono et al., 2018). However, our cashew plantations have less *DWB* than plantations in Cameroon (Victor et al., 2021). Variations in total *DWB* values could result from the degree of disturbances within the studied forests (Baker et al., 2007). The higher *DWB* in EF is due to its old stand age, long-term accumulation of *DWB*, and absence of slash-and-burn practice as observed in RF and CP (van Galen et al., 2019). In CP, some farmers periodically cut and burn dead branches of cashew trees to promote growth.

### 4.6 *LAI* and *fPAR*

In our study, canopy leaf area index in evergreen forests surpasses that of dry evergreen forests in Kampong Thom, Cambodia, while regrowth forests lie between those of 18–35-year tropical secondary forests in Costa Rica; however, cashew plantations exceed reported values in India (Ito et al., 2007; Clark et al., 2021; Kumaresh et al., 2023). The $LAI_C$ difference between the forests (EF and RF) and CP was significant due to CP management practices, resulting in a thin canopy with low $LAI_C$. In

contrast, natural forests with their densely developed canopy have a high $LAI_C$. Additionally, $LAI_C$ phenology followed the rainy and dry seasons, with peak values during the rainy season and low values during the dry season (Ito et al., 2007). During the dry season, reduced rainfall leads to less water availability for plant growth, causing plants to adapt to water stress by shedding their leaves, resulting in low $LAI_C$ in the ecosystem (Maréchaux et al., 2018). The comparison between $LAI_C$ and $LAI_T$ of EF and RF with previous studies is presented in Table S10.3.

Our mean fraction of photosynthetically active radiation for EF and RF marginally exceeded the global range for broadleaf forests and the monthly range observed in the Amazon tropical forest in Santarém, Brazil (Senna et al., 2005; Pastorello et al., 2020). The *fPAR* for CP, on the other hand, is within the range values reported for broadleaf crops (Xiao et al., 2015). Despite annual variations in $LAI_C$ (24 % for EF, 32 % for RF, 29 % for CP) and incoming solar irradiance, *fPAR* remained remarkably stable throughout the year in the forest ecosystems (EF and RF, Fig. 7). This stability can be attributed to the exponential



relationship between *fPAR* and *LAI*, which typically saturates at *LAI* above 3 (Dawson et al., 2003). Our recorded lowest *LAI*
for EF and RF was 3.48, likely contributing to this saturation and explaining the lack of phenology displayed in *fPAR*. The
exclusion of reflected *PAR* above the canopy in the *fPAR* estimation may also contribute to the stability; however, previous
studies have shown that the difference between intercepted (what we measured) and absorbed *PAR* (including the reflected
component) is minimal (Olofsson and Eklundh, 2007).

**4.7 $AGB_h$ relationships with $LAI_T$, $SLA_{cwm}$ and $S_R$**

Exploring the relationship between aboveground biomass and key ecosystem characteristics such as leaf area index, specific
leaf area, and species richness is vital for comprehending the complexity of ecosystem dynamics and informing ecosystem
modelling. We observed a strong positive relationship between $LAI_T$ and $AGB_h$, supporting prior findings (He et al., 2021;
Zhao et al., 2021). Higher $LAI_T$ enhances light interception and results in higher biomass. Elevated $AGB_h$ levels stimulate $LAI_T$
expansion by providing resources for robust leaf growth, leading to a denser canopy and greater leaf coverage. Similarly, our
findings support a positive relationship between $SLA_{cwm}$ and $AGB_h$ (Finegan et al., 2015; Ali et al., 2017; Gao et al., 2021).
Higher $SLA_{cwm}$ values indicate a plant community with improved photosynthetic capacity, nutrient uptake, and leaf turnover,
which is essential for nutrient cycling (Reich et al., 1991). An increase in $AGB_h$ has a reinforced effect on $SLA_{cwm}$ values,
suggesting enrichment of the soil nutrient pool and providing structural support for plant growth. This influences light
availability and competition dynamics, affecting leaf morphology and $SLA_{cwm}$. Furthermore, the positive relationship between
$AGB_h$ and $S_R$ is widely observed and explained by the niche complementarity hypothesis (Waide et al., 1999; Jactel et al.,
2018; Steur et al., 2022). This concept suggests that an ecosystem with high species diversity has a greater variation in
functional traits and resource-use strategies, lowering competition for scarce resources, and thus promoting productivity
(Tilman et al., 1997). In return, an increase in $AGB_h$ fosters the coexistence of diverse species by providing more available
resources and habitat complexity in an ecosystem, thereby increasing species richness.

**5 Conclusions**

Land use and land cover change is one of the most severe environmental challenges within the Earth system. In the context of
tackling current global environmental challenges, field observations are necessary to assess the dynamic responses of
ecosystems to changing environmental conditions on fine spatial and temporal scales. Especially Southeast Asia, renowned
for its biodiversity richness, suffers from a scarcity of integrated datasets that encompass a broad spectrum of ecosystem
characteristics across different land-cover classes. Here we present the first data of a newly established field site in a tropical
forest region of Southeast Asia (the Kulen National Park, Cambodia), where we started monitoring ecosystem characteristics
of land-cover classes with various anthropogenic pressures (pristine evergreen forests, regrowth forests, and cashew
plantations). We thereafter used the observed ecosystem characteristics for the land-cover classes with various anthropogenic
pressures, to provide a comprehensive analysis of changes in ecosystem characteristics between these classes. Our results





highlight a substantial reduction in soil water content, species diversity, leaf functions, stand structural complexity, aboveground biomass, deadwood, leaf area index, and fraction of photosynthetically active radiation absorbed by the tree canopy, in the land-cover classes affected by the anthropogenic land cover conversion. We further demonstrate the utility of our novel dataset for improving aboveground biomass estimation through the application of an allometric function based on

locally specific wood density and the *DBH-H* relationship. This approach has great potential for improving carbon stock estimations and promoting informed forest management practices. Moreover, our analysis of relationships between leaf area index, specific leaf area, species richness and aboveground biomass, underlines profound impact land-cover change has on ecosystem productivity and functioning in these tropical forest regions. We further expect that the dissemination of our datasets will contribute valuable insights for advancing the understanding of tropical forest ecosystems in Southeast Asia, support

research, and promote sustainable forest management under global environmental challenges.

## Appendix A

**Table A1. Estimated lying deadwood biomass (Mg ha$^{-1}$), standing deadwood biomass (Mg ha$^{-1}$), and total deadwood biomass (Mg ha$^{-1}$) by different land-cover classes in Kulen. Mean ± SD is a mean plus or minus a standard deviation.**

| Land cover | Lying deadwood biomass (Mg ha$^{-1}$) | | Standing deadwood biomass (Mg ha$^{-1}$) | | Total deadwood biomass (Mg ha$^{-1}$) | |
|---|---|---|---|---|---|---|
| | Mean ± SD | Range | Mean ± SD | Range | Mean ± SD | Range |
| EF (n = 3) | 17.74 ± 19.93 | 1.64–40.03 | 9.74 ± 8.49 | 0–15.56 | 27.48 ± 12.37 | 15.31–40.03 |
| RF (n = 3) | 3.65 ± 5.32 | 0.48–9.79 | 1.16 ± 1.66 | 0–3.06 | 4.81 ± 6.97 | 0.48–12.85 |
| CP (n = 3) | 0.40 ± 0.19 | 0.28–0.62 | 0 | 0 | 0.40 ± 0.19 | 0.28–0.62 |

**Table A2. Comparing estimated aboveground biomass (*AGB*, Mg ha$^{-1}$) in evergreen forests (EF) using adopted allometric equations (*AGB*$_f$), diameter at breast height (*DBH*) and tree height (*H*) power-law relationship (*AGB*$_h$), and previous *AGB* reported in previous studies. Mean ± SD is a mean plus or minus a standard deviation.**

| No. | Region | Vegetation type | *AGB* (Mg ha$^{-1}$) | | References |
|---|---|---|---|---|---|
| | | | Mean ± SD | Range | |
| 1 | Kulen, Cambodia | Tropical evergreen forest | 311.66 ± 183.88 | 147.53–510.57 | *AGB*$_h$ in this study |
| 2 | Kulen, Cambodia | Tropical evergreen forest | 238.53 ± 92.41 | 161.83–341.13 | *AGB*$_f$ in this study |
| 3 | Global | Tropical forest | 379.02 ± 187.40 | 230.58–589.58 | Chave et al. (2014) |
| 4 | Gia Lai, Vietnam | Tropical evergreen forest | 273.24 ± 112.22 | 189.53–400.76 | Nam et al. (2016) |
| 5 | Mondulkiri, Cambodia | Tropical moist evergreen forest | 333.00 ± 137.00 | 78.00–837.00 | Sola et al., (2014) |
| 6 | Borneo (Brunei, Malaysia, Indonesia) | Tropical lowland evergreen forest | 458.16 ± 123.62 | 196.30–778.50 | Slik et al. (2010)) |



| No. | Region | Vegetation type | AGB | Range | References |
|-----|--------|-----------------|-----|-------|------------|
| 7 | Thanh Hoa, Vietnam | Tropical evergreen broadleaf forest | 251.81 ± 125.43 | 40.88–543.88 | Nguyen and Kappas (2020) |
| 8 | Africa | Tropical evergreen forest | 429.00 | 114.00–749.00 | Lewis et al. (2013) |
| 9 | Cambodia | Evergreen forest | 243.00 ± 128.00 | 11.00–837.00 | Sola et al., (2014) |
| 10 | Kampong Thom, Cambodia | Evergreen forest | 294.00 ± 65.00 | 176.00–398.00 | Ota et al. (2015) |
| 11 | Vietnam | Tropical evergreen broadleaf forests in various ecoregions | 230.10 ± 8.60 | 199.00–320.20 | Van Do et al. (2020) |

**Table A3. Comparing estimated aboveground biomass (*AGB*, Mg ha⁻¹) in regrowth forests (RF) using adopted allometric equations (*AGB*f), diameter at breast height (*DBH*) and tree height (*H*) power-law relationship (*AGB*h), and previous *AGB* reported in previous studies. Mean ± SD is a mean plus or minus a standard deviation.**

| No. | Region | Vegetation type | *AGB* (Mg ha⁻¹) | | References |
|-----|--------|-----------------|-----------------|-------|------------|
| | | | Mean ± SD | Range | |
| 1 | Kulen, Cambodia | Natural regrowth evergreen forest | 54.19 ± 14.09 | 38.26–65.04 | *AGB*h in this study |
| 2 | Kulen, Cambodia | Natural regrowth evergreen forest | 41.66 ± 9.82 | 31.60–51.21 | *AGB*f in this study |
| 3 | Sumatra, Indonesia | Mixed secondary forest | 59.04 ± 17.15 | 39.26–69.79 | Ketterings et al. (2001) |
| 4 | Kampong Thom, Cambodia | Regrowth forest | 42.00 ± 21.00 | 22.00–90.00 | Ota et al. (2015) |
| 5 | Malaysia | Young forests aged 8.5–17 years | 63.60 ± 34.93 | 34.00–118.00 | Kho and Jepsen (2015) |

**Table A4. Comparing estimated aboveground biomass (*AGB*, Mg ha⁻¹) in cashew plantations (CP) using adopted allometric equations (*AGB*f), diameter at breast height (*DBH*) and tree height (*H*) power-law relationship (*AGB*h), and previous *AGB* reported in previous studies. Mean ± SD is a mean plus or minus a standard deviation.**

| No. | Region | Vegetation type | *AGB* (Mg ha⁻¹) | | References |
|-----|--------|-----------------|-----------------|-------|------------|
| | | | Mean ± SD | Range | |
| 1 | Kulen, Cambodia | Family-scale cashew plantation | 16.70 ± 4.80 | 11.23–20.23 | *AGB*h in this study |
| 2 | Kulen, Cambodia | Family-scale cashew plantation | 70.60 ± 22.01 | 46.16–88.87 | *AGB*f in this study |
| 3 | Benin | Cashew agroforestry farming | 18.07 ± 2.14 | - | Biah et al. (2019) |
| 4 | Guinean, Cote d'Ivoire | Cashew plantation | 13.78 ± 0.98 | - | Kanmegne Tamga et al. (2022) |
| 5 | Kampong Cham, Cambodia | Large-scale and intensively managed cashew plantation (10–16 years of age) | 104.30 ± 19.65 | 72.00–143.00 | Avtar et al. (2013) |



**Data availability**

All the collected data used in this study are publicly available via the links as follows:

1. The datasets of the forest inventory, leaf area index, and leaf functional traits across various land-cover classes are available at https://doi.org/10.5281/zenodo.10146582 (Sovann et al., 2024a).
2. The daily data, including *fPAR*, soil conditions, and meteorological conditions from April 10, 2022, to April 9, 2023, can be downloaded from https://doi.org/10.5281/zenodo.10159726 (Sovann et al., 2024b).
3. Future data from the field site will be uploaded to https://zenodo.org/communities/cambodia_ecosystem_data on a regular basis.

**Author contribution**

CS led field data collection, analysis, and manuscript writing. TT and SO contributed to conceptualization, manuscript review, editing, and supervision. SK and SS provided administrative support and supervised fieldwork in Cambodia. PV offered technical guidance and support in equipment installation and maintenance. SB managed field data collection. All authors contributed to editing the manuscript.

**Competing interests**

The authors declare no conflict of interest.

**Acknowledgements**

This work was supported by the Swedish International Development Cooperation Agency through the "Sweden-Royal University of Phnom Penh Bilateral program" (Contribution Number: 11599). Tagesson was additionally funded by the Swedish National Space Agency (SNSA Dnr: 2021-00144) and Formas (Dnr. 2021-00644; 2023-02436). The research
presented in this paper is a contribution to the Strategic Research Area "Biodiversity and Ecosystem Services in a Changing Climate", BECC, funded by the Swedish government.
We are grateful to the Ministry of Environment (Cambodia) and Siem Reap Provincial Administration for their grant permissions, administrative support, and accommodation during our fieldwork. A special note of thanks goes to Seng Saingheat for his dedication and leadership, along with his ranger colleagues, including Sou Sy, Let Chey, Khun Chi, Soun Sao, Kroem
Veng, Choun Choy, and Ti Has. Sincere appreciation to the research teams from the Royal University of Agriculture (Cambodia), namely Horn Sarun, Yorn Chomroeun, Sok Pheak, Sum Dara, and Long Sotheara, for their invaluable support





in forest inventory. We greatly appreciate Mot Ly and Chim Lychheng, as well as Rum Pheara, Svay Chanboth, and Mach Sokmean, for their support throughout our data collection journey.

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
