# Peer review of "Characteristics of ecosystems under various anthropogenic impacts in a tropical forest region of Southeast Asia"

_EGUsphere, 2024_

## Referee Comment (RC3)

*This tittle sounds too general*

**Characteristics of ecosystems under various anthropogenic impacts in a tropical forest region of Southeast Asia**

Chansopheaktra Sovann[1,2], Torbern Tagesson[1], Patrik Vestin[1], Sakada Sakhoeun[3], Soben Kim[4], Sothea Kok[2], Stefan Olin[1]

5 [1]Department of Physical Geography and Ecosystem Science, Lund University, Sölvegatan 12, S-223 62 Lund, Sweden
[2]Department of Environmental Science, Royal University of Phnom Penh, Phnom Penh, 120404, Cambodia
[3]Provincial Department of Environment, Ministry of Environment, Siem Reap, 171202, Cambodia
[4]Faculty of Forestry, Royal University of Agriculture, Phnom Penh, 120501, Cambodia

*Correspondence to*: Chansopheaktra Sovann (chansopheaktra.sovann@nateko.lu.se)

*which specific pressure? I would rather point out the importance of land-use/land cover change*

10 **Abstract.** Given the severe ==anthropogenic pressur==e on tropical forests and the high demand for field observations of ecosystem characteristics, it is crucial to collect such data ==both in pristine tropical forests and in the converted deforested== land-cover

*As you mentioned what was done in this research, land-cover change may be the main focus?*

[revised manuscript text omitted]

$$S_\text{H} = -\sum_{i=1}^{n} P_\text{i} \ln(P_\text{i}) \tag{2}$$

Where $S_\text{H}$ is Shannon-Wiener index (unitless), $P_\text{i}$ is a proportion of *i* species in a community (unitless), and *n* is the number of species in a plot (unitless). We calculated the $S_\text{R}$ and $S_\text{H}$ at the plot level and then averaged the values for each land-cover

165 class.

**2.3.2 Functional traits and diversity**

We computed the specific leaf area (*SLA*) for each of the 453 leaf samples as the ratio of leaf area to leaf dry mass. Likewise, leaf dry matter content (*LDMC*) was calculated by the ratio of dry leaf mass to fresh leaf mass (Garnier et al., 2001; Akram et al., 2023). We estimated the trait community-weighted means and standard deviations of $SLA_\text{cwm}$, $LDMC_\text{cwm}$, and $Chl_\text{cwm}$ to

170 represent ecosystem functions and their diversity at the land-cover level (Garnier et al., 2004; Leoni et al., 2009; Wang et al., 2020) with:

$$T_\text{cwm} = \frac{\sum_{i=1}^{n} W_\text{i} T_\text{i}}{\sum_{i=1}^{n} W_\text{i}} \tag{3}$$

[revised manuscript text omitted]

---

## Author Comment (AC3)

**Land-cover change alters stand structure, species diversity, leaf functional traits, and soil conditions in Cambodian tropical forests**

[revised manuscript text omitted]

45  services and goods (Tito et al., 2022). While the ecological consequences of forest conversion are broadly recognized, relatively few studies have comprehensively examined how transitions from primary to secondary forests and plantations influence multiple ecosystem characteristics, particularly through detailed field-based observations that may inform our understanding of ecosystem functioning.

Tropical forests demonstrate remarkable ecological complexity, with high diversity in stand structure, species composition,

50  and functional traits shaped by heterogeneous environments and varied disturbance histories (Manuel Villa et al., 2020). This complexity leads to highly site-specific and often inconsistent ecosystem characteristics and responses, making it difficult to generalize the impacts of land-cover change across regions (Wang et al., 2022). A parallel challenge in tropical forest research is the accurate estimation of aboveground biomass ($AGB$), a key metric for assessing carbon stocks. Most studies rely on generalized allometric models developed under different ecological conditions, assuming similarity in forest structure,

55  composition, and wood density, which these assumptions rarely hold in structurally diverse tropical forests (Vieilledent et al., 2012). These limitations introduce substantial uncertainty when models are applied across sites (Ketterings et al., 2001b). The lack of locally calibrated relationships between diameter at breast height ($DBH$) and tree height ($H$), wood density ($WD$), and the scarcity of direct destructive sampling further contribute to estimation errors, highlighting the need for site-specific approaches that reflect local variation in forest structure and composition.

60  In the context of tackling the current challenges of global land cover change, it is necessary to conduct field observations in order to investigate the responses of ecosystems to changing environmental conditions on fine spatial and temporal scales.

Field observations of key ecosystem characteristics such as forest inventory, leaf functional traits, leaf area index (*LAI*), fraction of photosynthetically active radiation (*fPAR*), and soil conditions provide crucial insights into ecosystem functions and services, including vegetation productivity, carbon sequestration, hydrological cycle, ecosystem stability and resilience to

65  disturbances, nutrient reservoir capacity, and the abundance of habitats of organisms (Naeem et al., 1994; Hector, 1998; Cardinale et al., 2012; Chen et al., 2016; Liang et al., 2016; Parisi et al., 2018b; Woodall et al., 2020). In addition, the field data on leaf functional traits, *LAI*, and *fPAR* are important for the parameterization and evaluation of remote sensing products and dynamic vegetation models, essential for modelling and upscaling ecosystem responses to anthropogenic disturbances and climate change (Feng et al., 2018; Fang et al., 2019; Pei et al., 2022). Recognizing the significant role and high demand for

70  field observations of ecosystem characteristics, open data repositories such as FLUXNET, ICOS Carbon Portal, SpecNet, and the TRY database have been established to facilitate data sharing (Gamon et al., 2010; Kattge et al., 2020; Pastorello et al., 2020). Despite those global initiatives, observed data from tropical forests that support multi-class, pairwise comparisons for capturing ecosystem changes across gradients of forest degradation and land-use conversion remain limited. This gap is especially critical in Southeast Asia, where rapid forest-to-agriculture transitions threaten key ecosystem functions, and

75  understanding these ecosystem changes is essential for informing evidence-based conservation and restoration strategies (Fan et al., 2024).

Within this context, Phnom Kulen National Park (Kulen) in Cambodia emerges as a critical landscape for investigating ecosystem responses to land cover change. Kulen is a hotspot for ecosystem service provisioning in Cambodia, mainly for water supply, potential carbon sink, and cultural services (Jacobson et al., 2022; Kim et al., 2023). It is the origin of the Khmer

80  Empire and contains numerous archaeological sites. The stream water from the mountain is not only used to support local livelihoods in water supply and irrigation downstream (Somaly et al., 2020). It is also the primary water source to recharge surface water and groundwater aquifers in the Angkor Wat, UNESCO World Heritage Site. Hence, the area is of high importance to ensure that the temples' foundations remain stable and maintain their surrounding forest ecosystem (Hang et al., 2016). However, previous studies revealed that the forestland in and around Kulen has been disturbed, largely driven by

85  agricultural expansion, particularly the spread of cashew plantations (Chim et al., 2019; Sovann et al., 2025).

Given increasing concerns about land use and land cover change threatening the high-value ecosystem functions of tropical forests like those in Kulen, our study aims to gain insight into the impact of land-cover conversion on key ecosystem characteristics. Specifically, our first objective is to assess the differences in stand structure, species diversity, leaf functional traits, and soil conditions between pristine tropical forests and the land cover the deforested regions are converted into

90  (regrowth forests and cashew plantations). We hypothesise that there will be a reduction in stand structural complexity, species composition, and leaf functional traits, and a marked change in soil conditions. Additionally, our second objective is to evaluate the feasibility of locally updating aboveground biomass estimates by applying power-law functions derived from site-specific relationships between diameter at breast height and tree height, with the hypothesis that locally calibrated *DBH-H* relationships will have a substantial effect on estimated aboveground biomass compared to regional or generalized allometric models. To

test these hypotheses, we will analyse a novel in situ dataset collected from pristine tropical forests, regrowth forests, and cashew plantations at a newly established ecosystem monitoring site in Phnom Kulen National Park, Cambodia.

**2 Materials and Methods**

**2.1 Study area and selection of plots**

The selected study area is the Phnom Kulen National Park located in the Siem Reap Province in north-west Cambodia (Fig. 1). It covers 37,380 ha predominantly on Jurassic-Cretaceous sandstone plateaus with the highest peak of 496 m (Matschullat, 2014; Geissler et al., 2019). In 2021, 72 % of Phnom Kulen National Park was forested, dominated by nearly intact tropical evergreen forests (EF) (30 %) and forests that regrow naturally after clear-cutting (RF) (7 %). The remaining 35 % of forest cover consisted of semi-evergreen, deciduous, and bamboo stands. Non-forest areas were dominated by household-scale cashew plantations (CP) (15 %), with the remaining 13 % consisting of croplands, paddy fields, settlements, and tree and rubber plantations (Sovann et al., 2025).

[Figure]

**Figure 1. The locations of the nine forest inventory plots and the meteorological station in the Phnom Kulen National Park, Cambodia. Note: the background land cover 2021 was derived from Sovann et al. (2025).**

Nine forest inventory plots were established in Kulen in December 2020, three within each of the EF, RF, and CP land-cover classes (Fig. 1, Table 1, Fig. S1.1), with a minimum separation of 250 meters to capture stand structure variation for each land-cover class. The EF plots represented tropical evergreen forests with no clear-cut history. The RF plots were dominated by at least 10-year-old natural regrowth forests, RF1 was clear-cut in 2009, while RF2 and RF3 experienced timber harvesting, burning, and fuelwood collection from 2006 to 2013. The CP plots were permanent rainfed cashew plantations, with cashew trees planted in 2013 in CP1 and in 2012 for the other two.

**Table 1. Characteristics of the forest inventory plots in Phnom Kulen National Park. Note: EF = evergreen forests, RF = regrowth forests, and CP = cashew plantations. Data source: Soil type and geology data from Matschullat (2014). Disturbance history information is obtained from field observation, discussion with local people, and combining with the Global Forest Change dataset of Hansen et al. (2013) and LandTrendr Pixel Time Series Plotter tool of Kennedy et al. (2018).**

| Plot ID | Latitude, Longitude | Elevation (m) | Soil type | Disturbance history |
|---------|---------------------|---------------|-----------|---------------------|
| EF1 | N 13° 34' 12.4680" E 104° 7' 18.6096" | 331 | Acid Lithosols | No clear-cut history; affected by wind disturbance and human collection of wild honey, lychee, and other wild fruits in 2006, 2012, and 2014. Fewer large tree stands and lower vegetation cover density compared to EF2 and EF3. |
| EF2 | N 13° 34' 25.3452" E 104° 7' 20.2872" | 349 | Acid Lithosols | No clear-cut history; past disturbances include cutting one lychee tree for fruit harvesting in 2022. |
| EF3 | N 13° 34' 35.0508" E 104° 7' 20.6148" | 339 | Acid Lithosols | No clear-cut history; a wind-driven disturbance occurred in 2023. |
| RF1 | N 13° 33' 42.6132" E 104° 8' 1.2408" | 331 | Red-yellow podzols | Evergreen forest clear-cut in 2009. |
| RF2 | N 13° 36' 15.6924" E 104° 7' 48.8928" | 371 | Acid Lithosols | Timber harvesting, burning, and fuelwood collection of an evergreen forest from 2006 to 2013. |
| RF3 | N 13° 37' 0.3612" E 104° 7' 41.358" | 401 | Acid Lithosols | Timber harvesting, burning, and fuelwood collection of an evergreen forest from 2006 to 2013. |
| CP1 | N 13° 32' 18.8988" E 104° 12' 12.5568" | 429 | Red-yellow podzols | Cashew plantation established in 2013. |
| CP2 | N 13° 32' 29.3100" E 104° 12' 13.0284" | 422 | Red-yellow podzols | Cashew plantation established in 2012. |
| CP3 | N 13° 32' 50.1864" E 104° 12' 13.1544" | 430 | Red-yellow podzols | Cashew plantation established in 2012. |

[revised manuscript text omitted]

Soil conditions varied significantly among land-cover classes (ANOVA and Tukey HSD, p-value < 0.001). Annual daily mean soil temperature was highest in CP (25.8 °C), exceeding values in EF (24.3 °C) and RF (24.2 °C). In contrast, annual daily mean soil water content was lowest in RF (0.14 m$^3$ m$^{-3}$) compared to EF (0.23 m$^3$ m$^{-3}$) and CP (0.21 m$^3$ m$^{-3}$) (Table 3). Annual daily mean soil electrical conductivity was highest in EF (0.039 dS m$^{-1}$), followed by RF (0.032 dS m$^{-1}$) and CP (0.025 dS m$^{-1}$). Overall, daily mean values across land-cover classes ranged between 0.14–0.23 m³ m⁻³ for *SWC*, 24.2–25.8 °C for *Ts*, and 0.025–0.039 dS m⁻¹ for *ECs* (measured at 20 cm depth, Fig. 2g–i).

[Figure]

**Figure 2. The meteorological conditions at Kulen meteorological station (a–f), and soil conditions at each land-cover class (g–i) from April 10, 2022, to April 9, 2023. (a) Daily mean air temperature ($T_{air}$, °C), (b) daily total precipitation ($P$, mm), (c) daily mean global radiation ($Rg$, W m$^{-2}$), (d) daily mean relative humidity ($RH$, %), (e) daily mean vapour pressure deficit ($VPD$, kPa), and (f) daily mean wind speed ($WS$, m s$^{-1}$), (g) daily mean soil water content ($SWC$, m$^3$ m$^{-3}$), (h) daily mean soil temperature ($Ts$, °C), (i) daily mean soil saturation extraction electrical conductivity ($ECs$, dS m$^{-1}$). The vertical dashed line region in all the plots highlighted the rainy season period in Cambodia from May to October. The grey-shaded regions around the mean in (a), (d), (e), and (f) represent the 95 % confidence interval (using a standard deviation) from the daily mean, whereas the blue horizontal dashed line represents**

the yearly mean, the brown horizontal dashed line represents the yearly median, and the black horizontal dotted line represents a yearly standard deviation (see Table S2.1 and Fig. S2.2 present the Kulen meteorological station's annual and monthly meteorological data. Figs. S3.1–S3.3 shows monthly mean soil conditions by land-cover class, and Fig. S4.1 depicts correlations between meteorological and soil conditions).

**3.2 Species diversity**

A total of 343 observations (292 trees and 51 seedlings) from 47 woody species (including 13 seedling species) and 32 families (including seven seedling families) were identified from the nine plots (Table S5.1). No statistical test of significance of differences in species diversity among land cover classes was possible due to too few sampled plots. However, species diversity declined markedly from evergreen forests to regrowth forests and was lowest in cashew plantations, as reflected in both species richness and in the Shannon-Wiener index. The average $S_R$ per plot was 17 in EF, 13 in RF, and only 4 in CP. Similarly, the $S_H$ was highest in EF (2.48 ± 0.33), intermediate in RF (1.97 ± 0.45), and lowest in CP (0.61 ± 0.46), with individual plot values ranging from 0.31 (CP2) to 2.68 (EF1) (Table S5.2). Species composition was more evenly distributed in EF and RF, but naturally strongly dominated by a single species in CP. In EF, the top five most abundant species, *Mesua ferrea* (n = 18), *Diospyros bejaudii* (n = 12), *Litchi chinensis* (n = 11), *Vatica odorata* (n = 11), and *Hydnocarpus annamensis* (n = 8), accounted for 46 % of the individuals. In RF, *Vatica odorata* (n = 54), *Nephelium hypoleucum* (n = 14), *Benkara fasciculata* (n = 12), *Garcinia oliveri* (n = 12), and *Mesua ferrea* (n = 5) made up 61 %. In contrast, CP was dominated by *Anacardium occidentale* (n = 46), which was the only tree species observed excluding seedlings. Additional seedling species in CP included *Strychnos axillaris* (n = 3), *Nephelium hypoleucum* (n = 1), *Melodorum fruticosum* (n = 1), *Maclura cochinchinensis* (n = 1), and *Catunaregam tomentosa* (n = 1). Furthermore, fast-growth species, as described by Ha (2015) ($WD < 0.6$ g cm$^{-3}$), accounted for 40 % of EF and 44 % of RF of their total species composition.

**Table 3. Mean values and statistics of ecosystem characteristics in the different land-cover classes.**

[revised manuscript text omitted]

Across land-cover classes, mean $SLA_{cwm}$ and $Chl_{cwm}$ decreased from EF to RF to CP. $SLA_{cwm}$ and $Chl_{cwm}$ were highest in EF ($18.18 \pm 2.86$ m$^2$ kg$^{-1}$ and $9.14 \pm 3.45$ mg g$^{-1}$) followed by RF ($14.87 \pm 2.06$ m$^2$ kg$^{-1}$ and $7.56 \pm 2.03$ mg g$^{-1}$) and CP ($11.99 \pm 1.45$ m$^2$ kg$^{-1}$ and $4.99 \pm 0.66$ mg g$^{-1}$). Both traits showed statistically significant differences across land covers (ANOVA p-value $< 0.002$ for $SLA_{cwm}$, p-value $< 0.018$ for $Chl_{cwm}$). In contrast, $LDMC_{cwm}$ did not differ substantially among land-cover classes (p-value $= 0.51$), with CP having the highest value ($407.64 \pm 21.68$ mg g$^{-1}$), followed by EF ($398.43 \pm 72.24$ mg g$^{-1}$) and RF ($370.13 \pm 94.97$ mg g$^{-1}$). See Table S6.4 for data sources and shared percentages of species trait values used to compute $SLA_{cwm}$, $Chl_{cwm}$, and $LDMC_{cwm}$.

**3.4 Stand structure attributes**

**DBH* and tree height**

Land-cover conversion reduces both the mean and variability of tree diameter and height, indicating a loss of structural complexity in human-disturbed ecosystems. Structural measurements of 292 woody trees across three land-cover classes showed that EF had the highest structural complexity, with the highest mean and variability in *DBH* ($18.0 \pm 20.1$ cm) and tree height ($17.0 \pm 13.3$ m), including the largest individuals ($DBH = 102.3$ cm, $H = 52.0$ m, Fig. S7.1). However, RF and CP had substantially lower means and variability in these variables, suggesting reduced structural complexity after forest conversion. While both RF and CP had similar heights (RF: $7.4 \pm 3.8$ m, CP: $6.3 \pm 1.0$ m), CP had a significantly greater *DBH* (CP: $13.0 \pm 3.9$ cm, RF: $5.8 \pm 4.3$ cm). The results of the ANOVA and Tukey HSD tests confirmed significant differences in *DBH* and height among land covers (p-value $< 0.001$), except for CP and EF for *DBH* and CP and RF for height (Table 3).

**Aboveground and deadwood biomass**

Land-cover conversion from EF to RF and CP resulted in a substantial decline in both aboveground and deadwood biomass. The mean $AGB_f$ estimated using the generic allometric function dropped sharply from $239 \pm 92$ Mg ha$^{-1}$ in EF to $42 \pm 10$ Mg ha$^{-1}$ in RF and $71 \pm 22$ Mg ha$^{-1}$ in CP. Similarly, the mean total *DWB* declined from $27.5 \pm 12.4$ Mg ha$^{-1}$ in EF, $4.8 \pm 7.0$ Mg ha$^{-1}$ in RF, and $0.4 \pm 0.2$ Mg ha$^{-1}$ in CP. See Table A1 for the contribution of lying and standing *DWB* to total *DWB*.

**Stem density and basal area**

Changes in land cover strongly influenced stem density, basal area, and the distribution of aboveground biomass across *DBH*
335    classes (Fig. 3). RF exhibited twice the stem density (*DBH* > 5 cm) per hectare compared to EF and CP, driven largely by a
high proportion of smaller trees in the 5–15 cm *DBH* class. Despite having a lower mean *DBH*, RF had a higher basal area
($17.0 \pm 5.4$ m² ha⁻¹) than CP ($11.6 \pm 3.5$ m² ha⁻¹). Interestingly, in EF, only 5 % of the stems with a *DBH* > 30 cm contributed
to approximately 65 % of the total *AGB*f. In contrast, the main *DBH* class contributing to the *AGB*f in RF and CP was 5–15
cm, accounting for 57 % and 76 % of the total *AGB*f in RF and in CP, respectively. Refer to Supplementary Table S7.1 for
340    shared stem density percentages per hectare across *DBH* classes, and Table S7.2 for shared percentages of *AGB*f categorized
by *DBH* class.

[Figure]

**Figure 3. Estimations per land-cover class of a mean number of stems per hectare (a), basal area (*BA*, m² ha⁻¹) (b), and mean
aboveground biomass separated by the different diameters at breast height (*DBH*) classes (c). In (c), the contribution of different
345    *DBH* classes to the mean aboveground biomass estimated by the *AGB*f method was used in this calculation. The error bars in the
figure represent one standard deviation.**

*LAI* **and** *fPAR*

The mean total leaf area index values were $6.16 \pm 0.67$ m² m⁻² for EF, $5.57 \pm 0.76$ m² m⁻² for RF, and $3.07 \pm 0.61$ m² m⁻² for
CP. The mean canopy *LAI* values were $4.62 \pm 0.5$ m² m⁻² for EF, $4.66 \pm 0.70$ m² m⁻² for RF, and $2.52 \pm 0.42$ m² m⁻² for CP.
350    The ANOVA analysis revealed a significant difference in mean *LAI*T and mean *LAI*C among the three land-cover classes, while
the Tukey HSD test did not find a significant difference in mean *LAI*T and mean *LAI*C between EF and RF (Table 3). The

phenology of both $LAI_T$ and $LAI_C$ revealed a similar pattern in EF and RF, with peak and base values in June and March, respectively (Fig. 4a–b, Table S7.3). The $LAI_T$ and $LAI_C$ patterns for CP resembled those of EF and RF but also had a strong decrease in April. Furthermore, the understory LAI ($LAI_U$; the difference between $LAI_T$ and $LAI_C$) for the various land-cover

355 classes indicates that the ground vegetation highly contributes to $LAI_T$ for EF and RF, while the contribution was minor for CP (Fig. 4c). In particular, the $LAI_U$ mean values within a year were approximately $1.54 \pm 0.57$ $m^2$ $m^{-2}$ for EF (25 %), $0.91 \pm 0.36$ $m^2$ $m^{-2}$ for RF (16 %), and $0.55 \pm 0.39$ $m^2$ $m^{-2}$ for CP (18 %). A general trend of high contribution $LAI_U$ to $LAI_T$ in June and low contribution in April was apparent for all land-cover classes.

[Figure]

**Figure 4. Total leaf area index ($LAI_T$, m² m⁻²), canopy leaf area index ($LAI_C$, m² m⁻²), and understory leaf area index ($LAI_U$, m² m⁻²), their variations across different months within a year for evergreen forests (EF), regrowth forests (RF), and cashew plantations (CP). The lines in the graph represent the connection between the mean $LAI$ values from one month to another.**

The observed mean annual $fPAR$ for EF, RF, and CP was high: $0.97 \pm 0.01$, $0.96 \pm 0.01$, and $0.76 \pm 0.06$, respectively (Table 3). The values of EF and RF exhibited minimal fluctuations throughout the year, whereas the $fPAR$ of CP ranged between 0.55

365 and 0.93 (Fig. 5). Like $LAI$, the annual mean $fPAR$ among EF, RF, and CP were statistically significantly different according to both the ANOVA test and Tukey HSD's tests.

[Figure]

**Figure 5. Daily mean global radiation (*Rg,* W m⁻²) (a) and daily mean *fPAR* for evergreen forests (EF) (b), regrowth forests (RF) (c), cashew plantations (CP) (d) from April 11, 2022, to April 9, 2023 at Kulen. The shaded area represents one standard deviation from the mean, computed using the ten *PAR* sensors installed in each land-cover class.**

**3.5 Estimated Aboveground biomass based on *DBH-H* relationship**

**DBH-H* relationship**

Land-cover change weakens tree allometry, reducing the consistency of *DBH-H* relationships in human-impact forest and agricultural ecosystems. Strong positive relationships between *DBH* and *H* were observed in both EF and RF. For EF, 92 % of the variation in *H* can be explained by the variation in *DBH*, whereas for RF and CP, it was 78 % and 51 %, respectively (Fig. 6, Table S7.4). The power-law relationships between *DBH* and *H* further indicated that the $K_1$ and $K_2$ values for EF and RF were similar, whereas the values for CP were much lower. For a plot-level analysis of relationships between ln(*DBH*) and ln(*H*), see Fig. S7.2 and Table S7.5.

[Figure]

$\hat{y} = 2.34\ x^{0.72}$, $R^2 = 0.92$, p-value < 0.001
$\hat{y} = 2.21\ x^{0.70}$, $R^2 = 0.78$, p-value < 0.001
$\hat{y} = 2.56\ x^{0.35}$, $R^2 = 0.51$, p-value < 0.001

Land cover • EF • RF • CP

**Figure 6. Relationship between diameter at breast height (*DBH*) (cm) and tree height (*H*) (m) for evergreen forests (EF), regrowth forests (RF), and cashew plantations (CP) in Kulen. Figure shows the derived power-law intercept ($K_1$) and slope ($K_2$) values for EF, RF, and CP.**

**Comparison of *AGB* estimation methods**

Our results indicate that locally calibrated *DBH-H* relationships and species-specific wood density substantially affected aboveground biomass estimates compared to generalized models (Fig. 7). The $AGB_{wd}$ method consistently produced higher values than $AGB_f$ across all land-cover classes, reflecting the influence of wood density and the dominance of high-density tree species at our study site. In EF and RF, where *DBH-H* relationships were strong, $AGB_h$ estimates were markedly higher than $AGB_f$ (EF: $239 \pm 92$ vs. $312 \pm 184$ Mg ha$^{-1}$, RF: $42 \pm 10$ vs. $54 \pm 14$ Mg ha$^{-1}$), consistent with plot-level regression results (Fig. S7.2, Table S7.5). In contrast, in CP, $AGB_h$ yielded much lower values than $AGB_f$ ($17 \pm 5$ vs. $71 \pm 22$ Mg ha$^{-1}$), highlighting the limited reliability of this method under weak *DBH-H* relationship conditions. The differences between $AGB_h$ and $AGB_f$ estimates across land covers are illustrated in 1:1 comparison plots and plot-level summaries (Figs. S7.3–S7.5).

[revised manuscript text omitted]

**4.4 Stand structure attributes**

*DBH* **and tree height**

Our findings confirm significant differences in mean *DBH* and tree height resulting from the conversion of pristine evergreen forests to young regrowth forests and cashew plantations following human disturbance. The observed reduction in large-

490    diameter and tall trees in regrowth forests and cashew plantations compared to the evergreen forests (Fig. 3) provides clear evidence of structural degradation, which negatively affects crucial key ecosystem functions such as carbon storage, nutrient cycling, and biodiversity (Díaz et al., 2007; Lutz et al., 2018; Thiel et al., 2021). Observed species in our evergreen forests, such as *Dipterocarpus costatus*, *Sandoricum indicum*, *Mesua ferrea*, *Nageia wallichiana*, and *Litchi chinensis* reach heights of 40–52 m, similar to those found in Cambodia's central evergreen forests (Theilade et al., 2022). Our mean *DBH* of evergreen

495    forests is comparable to mature tropical forests in Vietnam and falls within the pantropical range, while regrowth forests have a slightly higher mean *DBH* than tropical secondary forests in Sarawak, Malaysia (Brown, 1997; Kenzo et al., 2009; Yen and Cochard, 2017). In contrast, cashew plantations show a significantly lower mean *DBH* compared to older counterparts in Kampong Cham, Cambodia (Avtar et al., 2013).

**Aboveground and deadwood biomass**

500    Our results support that land-cover conversion reduces in aboveground and deadwood biomass in regrowth forests and cashew plantations compared to evergreen forests. The substantial decline in aboveground biomass following conversion from EF to RF or CP is primarily driven by historical human disturbance, particularly clear-cutting and the removal of large trees, as evidenced by reduced *DBH* and tree height in this study. Similarly, *DWB* decreased as EF were replaced by RF and CP, reflecting the impacts of land-cover change on forest biodiversity and ecosystem health. *DWB* is a key indicator of biodiversity

505    and ecosystem health, supporting various species and ecosystem processes like carbon and nitrogen cycling, soil fertility enhancement, pollination, and erosion control (Parisi et al., 2018a; Santopuoli et al., 2021; Tláskal et al., 2021). Variations in total *DWB* values could result from the degree of disturbances within the studied forests (Baker et al., 2007). The higher *DWB* in EF is due to its old stand age, long-term accumulation of *DWB*, and absence of slash-and-burn practice as observed in RF and CP (van Galen et al., 2019). In CP, some farmers periodically cut and burn dead branches of cashew trees to promote

510    growth. Consistent with these trends, *DWB* in our EF and RF was comparable to previous studies in Cambodia and Malaysia (Saner et al., 2012; Kiyono et al., 2018), whereas our CP has less *DWB* than plantations in Cameroon (Victor et al., 2021).

**Stem density and basal area**

[revised manuscript text omitted]

**Comparison of *AGB* estimation methods**

Our results suggest that locally calibrated *DBH-H* relationships and wood density substantially affect *AGB* estimates compared to generalized models, supporting the feasibility of site-specific calibration, particularly for natural forest ecosystems. Recent

studies have emphasized the significant uncertainty in estimating plot-level aboveground biomass when directly applying a generic $AGB$ allometric equation ($AGB_f$) due to variations in species composition and stand structure between the study site and the equation's origin (Feldpausch et al., 2011; Burt et al., 2020). To address this challenge, our study proposes an allometric approach ($AGB_h$) using local species-specific wood density and the $DBH$-$H$ relationship at the study site. This approach captures the unique characteristics of the site's species composition and stand structure (Ketterings et al., 2001a; Nyirambangutse et al., 2017). Our locally adopted $AGB_h$ method produced estimates ~ 30 % higher than the generic $AGB_f$ for both EF and RF (Table 3, Fig. 7b). This is likely due to the combined effects of higher mean wood density and a stronger $DBH$ relationship, resulting in a more pronounced exponential growth response in $AGB$ (Fig. 7a). Still, these ~ 30 % higher values align with the range reported in previous studies (Tables A2–A3). In contrast, in the CP case, our $AGB_h$ method produced estimates less than a quarter of the generic $AGB_f$ method. The reason is that the $AGB_h$ method is less reliable when a weak $DBH$-$H$ relationship is detected because it fails to accurately capture the overall tree size and volume. This is also reflected in the substantially larger uncertainty as indicated by the standardized errors of the parameters within the $DBH$-$H$ relationship (Table A4, Table S7.4). The substantial difference between $AGB_{wd}$ and $AGB_f$ is primarily due to the wood density values used: 0.45 g cm$^{-3}$ from Zanne et al. (2009) in this study versus 0.18 g cm$^{-3}$ in the original $AGB_f$ equation (Mlagalila, 2016), likely reflecting variation in cashew wood properties or wood density measurement protocols among the two studies. Despite clear differences in Fig. 7b, formal statistical comparisons were not conducted due to the limited number of plots per class (n = 3), which restricts statistical power. However, to fully validate the $AGB$ allometric equations, destructive field-observed data would be necessary. Therefore, future research should include direct field measurements of $AGB$ to more accurately validate the methods for these land-cover classes.

**4.6 $AGB_h$ relationships with $LAI_T$, $SLA_{cwm}$ and $S_R$**

Exploring the relationship between aboveground biomass and key ecosystem characteristics such as leaf area index, specific leaf area, and species richness is vital for comprehending the complexity of ecosystem dynamics and informing ecosystem modelling. We observed a strong positive relationship between $LAI_T$ and $AGB_h$, supporting prior findings (He et al., 2021; Zhao et al., 2021). Higher $LAI_T$ enhances light interception and results in higher biomass. Elevated $AGB_h$ levels stimulate $LAI_T$ expansion by providing resources for robust leaf growth, leading to a denser canopy and greater leaf coverage. Similarly, our findings support a positive relationship between $SLA_{cwm}$ and $AGB_h$ (Finegan et al., 2015; Ali et al., 2017; Gao et al., 2021). Higher $SLA_{cwm}$ values indicate a plant community with improved photosynthetic capacity, nutrient uptake, and leaf turnover, which is essential for nutrient cycling (Reich et al., 1991). An increase in $AGB_h$ has a reinforced effect on $SLA_{cwm}$ values, suggesting enrichment of the soil nutrient pool and providing structural support for plant growth. This influences light availability and competition dynamics, affecting leaf morphology and $SLA_{cwm}$. Furthermore, the positive relationship between $AGB_h$ and $S_R$ is widely observed and explained by the niche complementarity hypothesis (Waide et al., 1999; Jactel et al., 2018; Steur et al., 2022). This concept suggests that an ecosystem with high species diversity has a greater variation in functional traits and resource-use strategies, lowering competition for scarce resources, and thus promoting productivity

(Tilman et al., 1997). In return, an increase in $AGB_h$ fosters the coexistence of diverse species by providing more available resources and habitat complexity in an ecosystem, thereby increasing species richness.

**5 Conclusions**

In response to growing concerns over the ecological impacts of forest conversion in tropical Southeast Asia, we investigated how land-cover change from pristine evergreen forests to regrowth forests and cashew plantations alters stand structure, species diversity, functional traits, and soil conditions, and evaluated the feasibility of locally calibrated *DBH-H* allometries for improving aboveground biomass estimation. Our findings confirm our hypotheses that land-cover change reduces stand structural complexity, species composition, and leaf functional traits, and causes a substantial change in soil conditions. We further demonstrate the utility of our novel dataset for improving aboveground biomass estimation through the application of an allometric function based on locally specific wood density and the *DBH-H* relationship. This approach has great potential for improving carbon stock estimations and promoting informed forest management practices. However, as we lack direct destructive samples of aboveground biomass, we can neither reject nor support our second hypothesis that locally calibrated *DBH-H* relationships would substantially improve aboveground biomass estimates compared to generalized models. Moreover, our analysis of relationships between leaf area index, specific leaf area, species richness, and aboveground biomass, underlines land-cover change's profound impact on ecosystem productivity and functioning in these tropical forest regions. To strengthen and extend these findings, future studies should incorporate destructive sampling to validate our locally calibrated aboveground biomass allometric equations based on *DBH-H* relationships and wood density. Expanding field data collection by increasing the number and spatial distribution of plots across a broader range of land-use classes in tropical Southeast Asia and promoting open data sharing will be critical for improving our understanding of ecosystem responses to forest conversion and supporting sustainable forest management under global change in the region.

[revised manuscript text omitted]
705       Oliveira, R. C., Jr., Souza, C. M., Jr., Moura, N. G., Nunes, S. S., Siqueira, J. V., Pardini, R., Silveira, J. M., Vaz-de-Mello, F. Z., Veiga, R. C., Venturieri, A., and Gardner, T. A.: Anthropogenic Disturbance in Tropical Forests Can Double Biodiversity Loss from Deforestation, Nature, 535, 144-147, https://doi.org/10.1038/nature18326, 2016.

Bezerra, M. A., Lacerda, C. F. D., Gomes Filho, E., Abreu, C. E. B. D., and Prisco, J. T.: Physiology of Cashew Plants
710      Grown under Adverse Conditions, Brazilian Journal of Plant Physiology, 19, 449-461,
     https://doi.org/10.1590/s1677-04202007000400012, 2007.

Bhandari, S. K., Veneklaas, E. J., McCaw, L., Mazanec, R., Whitford, K., and Renton, M.: Effect of Thinning and Fertilizer
     on Growth and Allometry Of, Forest Ecology and Management, 479, 118594,
     https://doi.org/10.1016/j.foreco.2020.118594, 2021.

715 Biah, I., Guendehou, S., Goussanou, C., Kaire, M., and Sinsin, B.: Allometric Models for Estimating Biomass Stocks in
     Cashew (Linnaeus) Plantation in Benin, 2019.

Brown, S.: Estimating Biomass and Biomass Change of Tropical Forests: A Primer, Food and Agriculture Organization of
     the United Nations, 1997.

Burt, A., Calders, K., Cuni-Sanchez, A., Gómez-Dans, J., Lewis, P., Lewis, S. L., Malhi, Y., Phillips, O. L., and Disney, M.:
720      Assessment of Bias in Pan-Tropical Biomass Predictions, 3, https://doi.org/10.3389/ffgc.2020.00012, 2020.

Cardinale, B. J., Duffy, J. E., Gonzalez, A., Hooper, D. U., Perrings, C., Venail, P., Narwani, A., Mace, G. M., Tilman, D.,
     Wardle, D. A., Kinzig, A. P., Daily, G. C., Loreau, M., Grace, J. B., Larigauderie, A., Srivastava, D. S., and Naeem,
     S.: Biodiversity Loss and Its Impact on Humanity, Nature, 486, 59-67, https://doi.org/10.1038/nature11148, 2012.

Chave, J., Andalo, C., Brown, S., Cairns, M. A., Chambers, J. Q., Eamus, D., Folster, H., Fromard, F., Higuchi, N., Kira, T.,
725      Lescure, J. P., Nelson, B. W., Ogawa, H., Puig, H., Riera, B., and Yamakura, T.: Tree Allometry and Improved
     Estimation of Carbon Stocks and Balance in Tropical Forests, Oecologia, 145, 87-99,
     https://doi.org/10.1007/s00442-005-0100-x, 2005.

Chave, J., Rejou-Mechain, M., Burquez, A., Chidumayo, E., Colgan, M. S., Delitti, W. B., Duque, A., Eid, T., Fearnside, P.
     M., Goodman, R. C., Henry, M., Martinez-Yrizar, A., Mugasha, W. A., Muller-Landau, H. C., Mencuccini, M.,
730      Nelson, B. W., Ngomanda, A., Nogueira, E. M., Ortiz-Malavassi, E., Pelissier, R., Ploton, P., Ryan, C. M.,
     Saldarriaga, J. G., and Vieilledent, G.: Improved Allometric Models to Estimate the Aboveground Biomass of
     Tropical Trees, Glob Chang Biol, 20, 3177-3190, https://doi.org/10.1111/gcb.12629, 2014.

Chen, B. Q., Li, X. P., Xiao, X. M., Zhao, B., Dong, J. W., Kou, W. L., Qin, Y. W., Yang, C., Wu, Z. X., Sun, R., Lan, G.
     Y., and Xie, G. S.: Mapping Tropical Forests and Deciduous Rubber Plantations in Hainan Island, China by
735      Integrating Palsar 25-M and Multi-Temporal Landsat Images, International Journal of Applied Earth Observation
     and Geoinformation, 50, 117-130, https://doi.org/10.1016/j.jag.2016.03.011, 2016.

Chheng, K., Sasaki, N., Mizoue, N., Khorn, S., Kao, D., and Lowe, A.: Assessment of Carbon Stocks of Semi-Evergreen
     Forests in Cambodia, Global Ecology and Conservation, 5, 34-47, https://doi.org/10.1016/j.gecco.2015.11.007,
     2016.

740 Chim, K., Tunnicliffe, J., Shamseldin, A., and Ota, T.: Land Use Change Detection and Prediction in Upper Siem Reap
     River, Cambodia, Hydrology, 6, 64, https://doi.org/10.3390/hydrology6030064, 2019.

Chim, K., Tunnicliffe, J., Shamseldin, A., and Sarun, S.: Sustainable Water Management in the Angkor Temple Complex,
     Cambodia, Sn Applied Sciences, 3, https://doi.org/10.1007/s42452-020-04030-0, 2021.

Clark, D. B., Oberbauer, S. F., Clark, D. A., Ryan, M. G., and Dubayah, R. O.: Physical Structure and Biological
745      Composition of Canopies in Tropical Secondary and Old-Growth Forests, PLoS One, 16, e0256571,
     https://doi.org/10.1371/journal.pone.0256571, 2021.

Con, T. V., Thang, N. T., Ha, D. T. T., Khiem, C. C., Quy, T. H., Lam, V. T., Van Do, T., and Sato, T.: Relationship
     between Aboveground Biomass and Measures of Structure and Species Diversity in Tropical Forests of Vietnam,
     Forest Ecology and Management, 310, 213-218, https://doi.org/10.1016/j.foreco.2013.08.034, 2013.

750 Coste, S., Baraloto, C., Leroy, C., Marcon, É., Renaud, A., Richardson, A. D., Roggy, J. C., Schimann, H., Uddling, J., and
     Hérault, B.: Assessing Foliar Chlorophyll Contents with the Spad-502 Chlorophyll Meter: A Calibration Test with
     Thirteen Tree Species of Tropical Rainforest in French Guiana, Annals of Forest Science, 67, 607-607,
     https://doi.org/10.1051/forest/2010020, 2010.

Dawson, T. P., North, P. R. J., Plummer, S. E., and Curran, P. J.: Forest Ecosystem Chlorophyll Content: Implications for
755      Remotely Sensed Estimates of Net Primary Productivity, International Journal of Remote Sensing, 24, 611-617,
     https://doi.org/10.1080/01431160304984, 2003.

Deng, C., Zhang, S. G., Lu, Y. C., Froese, R. E., Ming, A. G., and Li, Q. F.: Thinning Effects on the Tree Height–Diameter
     Allometry of Masson Pine (Pinus Massoniana Lamb.), Forests, 10, 1129, https://doi.org/10.3390/f10121129, 2019.

Díaz, S., Lavorel, S., De Bello, F., Quétier, F., Grigulis, K., and Robson, T. M.: Incorporating Plant Functional Diversity Effects in Ecosystem Service Assessments, Proceedings of the National Academy of Sciences, 104, 20684-20689, https://doi.org/10.1073/pnas.0704716104, 2007.

Fan, F., Li, W., Feng, Z., and Yang, Y.: Combining Landscape Patterns and Ecosystem Services to Disclose Ecosystem Changes in Tropical Cropland-Forest Shifting Zones: Inspiration from Mainland Southeast Asia, Journal of Cleaner Production, 434, 140058, https://doi.org/https://doi.org/10.1016/j.jclepro.2023.140058, 2024.

Fang, H., Baret, F., Plummer, S., and Schaepman-Strub, G.: An Overview of Global Leaf Area Index (Lai): Methods, Products, Validation, and Applications, Reviews of Geophysics, 57, 739-799, https://doi.org/https://doi.org/10.1029/2018RG000608, 2019.

[revised manuscript text omitted]

Horel, Á., Zsigmond, T., Molnár, S., Zagyva, I., and Bakacsi, Z.: Long-Term Soil Water Content Dynamics under Different Land Uses in a Small Agricultural Catchment, Journal of Hydrology and Hydromechanics, 70, 284-294, https://doi.org/10.2478/johh-2022-0015, 2022.

Howell, S. R., Song, G.-Z. M., Chao, K.-J., Doley, D., and Camac, J.: Functional Evaluation of Height–Diameter
855 Relationships and Tree Development in an Australian Subtropical Rainforest, Australian Journal of Botany, 70, 158-173, https://doi.org/10.1071/bt21049, 2022.

Huxley, J.: Problems of Relative Growth, L. MacVeagh, The Dial Press, New York, 1932.

Ibrahim, H. M. and Alghamdi, A. G.: Effect of the Particle Size of Clinoptilolite Zeolite on Water Content and Soil Water Storage in a Loamy Sand Soil, 13, 607, 2021.

860    Icraf Database: http://db.worldagroforestry.org/, last access: 21 May 2022.

Ito, E., Khorn, S., Lim, S., Pol, S., Tith, B., Pith, P., Tani, A., Kanzaki, M., Kaneko, T., Okuda, Y., Kabeya, N., Nobuhiro, T., and Araki, M.: Comparison of the Leaf Area Index (Lai) of Two Types of Dipterocarp Forest on the West Bank of the Mekong River, Cambodia, Forest Environments in the Mekong River Basin, 214-+, https://doi.org/10.1007/978-4-431-46503-4_19, 2007.

865    Jacobson, C., Smith, J., Sou, S., Nielsen, C., and Hang, P.: Effective Water Management for Landscape Management in the Siem Reap Catchment, Cambodia, in: Biodiversity-Health-Sustainability Nexus in Socio-Ecological Production Landscapes and Seascapes (Sepls), Satoyama Initiative Thematic Review, 
[revised manuscript text omitted]

Vieilledent, G., Vaudry, R., Andriamanohisoa, S. F. D., Rakotonarivo, O. S., Randrianasolo, H. Z., Razafindrabe, H. N., Rakotoarivony, C. B., Ebeling, J., and Rasamoelina, M.: A Universal Approach to Estimate Biomass and Carbon Stock in Tropical Forests Using Generic Allometric Models, Ecological Applications, 22, 572-583,
1340     https://doi.org/https://doi.org/10.1890/11-0039.1, 2012.

Waide, R. B., Willig, M. R., Steiner, C. F., Mittelbach, G., Gough, L., Dodson, S. I., Juday, G. P., and Parmenter, R.: The Relationship between Productivity and Species Richness, Annual Review of Ecology and Systematics, 30, 257-300, https://doi.org/10.1146/annurev.ecolsys.30.1.257, 1999.

Wang, P., Li, R., Liu, D., and Wu, Y.: Dynamic Characteristics and Responses of Ecosystem Services under Land Use/Land
1345     Cover Change Scenarios in the Huangshui River Basin, China, Ecological Indicators, 144, 109539, https://doi.org/https://doi.org/10.1016/j.ecolind.2022.109539, 2022.

[revised manuscript text omitted]

---

## Author Response (AR1)

**Land-cover change alters stand structure, species diversity, leaf functional traits, and soil conditions in Cambodian tropical forests**

Chansopheaktra Sovann[1,2], Torbern Tagesson[1], Patrik Vestin[1], Sakada Sakhoeun[3], Soben Kim[4], Sothea Kok[2], Stefan Olin[1]

[1]Department of Physical Geography and Ecosystem Science, Lund University, Sölvegatan 12, 22362 Lund, Sweden
[2]Department of Environmental Science, Royal University of Phnom Penh, Phnom Penh, 120404, Cambodia
[3]Provincial Department of Environment, Ministry of Environment, Siem Reap, 171201, Cambodia
[4]Faculty of Forestry, Royal University of Agriculture, Phnom Penh, 120501, Cambodia

*Correspondence to*: Chansopheaktra Sovann (chansopheaktra.sovann@nateko.lu.se)

**Response to Referee's comments (RC1)**

**Manuscript DOI:** https://doi.org/10.5194/egusphere-2024-3784
**Comment's citation:** https://doi.org/10.5194/egusphere-2024-3784-RC1  posted on 20250519

**Comment 1.** I congratulate the authors on their work. The manuscript presents a broad assessment of forest structural characteristics across different land uses in Cambodia. However, the authors have not fully explored the potential of this extensive dataset.

For instance, the introduction should clearly justify the importance of sampling these variables, which range from functional traits to climatic factors. As it stands, the study is primarily **descriptive and lacks a hypothesis-driven approach.** Strengthening these aspects would significantly improve the manuscript.

**Response 1 (previously posted).** Thank you for your thoughtful feedback and for recognizing the value of our work. We appreciate your suggestion to strengthen the justification for sampling key ecosystem characteristics. In response, we have revised the third paragraph of the introduction to better highlight the significance of the collected variables, including forest inventory, leaf functional traits, leaf area index (*LAI*), fraction of photosynthetically active radiation (*fPAR*), and soil conditions, supported by additional references (Page 3, Lines 45–59).

We find it difficult to convert the study into a having a more hypothesis-driven approach, given that there are so many different variables that are measured. But, we fully agree that the aims and objectives should not be primarily descriptive, but to more focus on the analysis of impact of the land cover conversion. Hence, we have rewritten the fourth paragraph of the introduction to clearly articulate the study's objectives. Specifically, we now emphasize our primary aim of assessing differences in stand structure, species diversity, leaf functional traits, and soil conditions between pristine tropical forests and the land cover types resulting from deforestation (regrowth forests and cashew plantations). Our second objective was to analyse the relationships between the characteristics and how these are influenced by land cover conversions. Lastly, we highlight our final objective of presenting and sharing this unique dataset to contribute to broader ecosystem research (Page 3, Lines 60–68).

**Response 1 (updated).** We would like to update our response related to this comment, as this issue has been pointed out by all three reviewers.

We appreciate the reviewer's comments. We agree that a clearer articulation of scientific questions and hypotheses strengthens the manuscript's alignment with *Biogeosciences*' scope. We have therefore substantially revised the manuscript to frame it around explicit scientific objectives and testable hypotheses regarding the impacts of land-cover change on ecosystem characteristics. Specifically:

- We revised the research objectives to emphasize analytical aims and formulated hypotheses. We now explicitly hypothesize that land-cover change reduces stand structure, species diversity, and leaf functional traits, and marked changes in soil conditions. Additionally, we evaluate whether locally calibrated *DBH*–height relationships significantly improve *AGB* estimates compared to generalized models. (Lines 86–96)
- The Abstract has been revised to emphasize the analytical objectives rather than focusing on dataset description (Lines 15–18), and references to data-sharing links have been removed.
- The Introduction section was comprehensively rewritten to support the updated research objectives and hypotheses. (Lines 45–85)

- We revised each Results subsection to align with our revised objective, which emphasizes how land-cover change alters key ecosystem characteristics. Specifically, we revised the following sections:
    - Soil conditions (Lines 256–261)
    - Species diversity (Lines 277–289)
    - Leaf traits (Lines 311–317)
    - Stand structure attributes
        - *DBH* and tree height (Lines 320–327)
        - Aboveground and deadwood biomass (Lines 329–332)
        - Stem density and basal area (Lines 334–339, Fig. 3)
- We introduced a new main section, "Estimated Aboveground Biomass Based on DBH-Height Relationship," in both the Results and Discussion to support our second hypothesis of applying locally calibrated DBH–H relationships for improving aboveground biomass estimation. (Lines 371–391, 564)
- We also updated the Discussion sections to support the revised results above: section 4.2 species diversity (lines 437–442), 4.3 Leaf functional traits, 4.4 Stand structure attributes: DBH and tree height (Lines 488–495), Aboveground and deadwood biomass (lines 500–511)

**Comment 2.** Abstract: L16: "Reduction in ecosystem characteristics" is unclear—consider rephrasing.

Response 2 (previously posted). We agree. We have rephrased "Reductions in several ecosystem characteristics" to "differences in these ecosystem characteristics" (Page 1, Lines 16–18).

**Response 2 (updated).** The response remains unchanged; the updated text is now located at Lines 18–20.

**Comment 3.** Graphical Abstract: Avoid abbreviations without prior explanation (e.g., *fPAR*).

Response 3 (previously posted). Thank you for pointing this out. We have replaced "*fPAR*" with "fraction of photosynthetically active radiation" in the graphical abstract (Page 1, Line 26).

**Response 3 (updated).** The response remains unchanged; the updated text is now located at Line 27.

**Comment 4.** Introduction: L33: Consider citing Pan et al. (2024, Science) for a more up-to-date reference.

Response 4 (previously posted). Thank you for this suggestion. We have updated the reference to Pan et al., 2024 and revised the text from "store approximately 60 % of the global terrestrial biomass (Pan et al., 2013)" to "account for approximately 70 % of the global forest gross carbon sink (Pan et al., 2024)" (Page 2, Lines 30–31).

**Response 4 (updated).** The response remains unchanged; the updated text is now located at Lines 31–33.

**Comment 5.** Methods: L75: Clarify the source of the carbon sink information.

Response 5 (previously posted). We have added a new reference (Kim et al., 2023) to support our statement that Kulen is a potential carbon sink (Pages 3–4, Lines 73–74).

**Response 5 (updated).** The response remains unchanged; however, we have moved these sentences from the Study Area section to the Introduction section to respond to Comment 9 from Reviewer 3. (Lines 77–85)

**Comment 6.** L94: The characterization of this area would be more useful in the main file rather than as supplementary material.

Response 6 (previously posted). We agree. Table S1.1 has been modified and moved to the main text as Table 1 (Pages 5–6, Line 92), and all table numbers in the main text and supplementary material have been updated accordingly.

**Response 6 (updated).** The response remains unchanged; the updated text is now located at Line 115.

**Comment 7.** L158–L162: Since plots have different sampling efforts, use rarefied richness instead of raw species number.

Response 7 (previously posted). We disagree with the comment that our plots have different sampling efforts. and prefer to use raw species numbers instead of rarefied richness. Rarefaction methods are typically applied when there are differences in sampling effort (Staudhammer et al., 2018). Our forest inventory for each land cover consisted of three plots with the same plot size, and all inventories were conducted within the same week in December 2020 following the same protocol of the National Forest Inventory of Cambodia (Page 4, Line 86; Page 6, Line 98). Therefore, we believe that raw species counts are suitable and more straightforward comparisons between land covers.

**Response 7 (updated).** The response remains unchanged; the updated text is now located at Line 109 for the date when conducting the forest inventory and at Lines 122–124 for the inventory plot size.

**Comment 8.** L175: I recommend using genus/family values before the mean plot value.

Response 8 (previously posted). Thanks for this great suggestion and have revised the methods accordingly (Page 8–9, Lines 176–178). The results in both the main text and supplementary material have been updated to reflect this methodological change. However, this adjustment had a minimal effect on the overall analysis, as only four species (*Agave sisalana, Dialium cochinchinense, Syzygium formosanum,* and *Madhuca elliptica*) had missing trait values.

**Response 8 (updated).** The response remains unchanged; the updated text is now located at Lines 198–200.

**References**

Kim, S., Horn, S., Sok, P., Sien, T., and Yorn, C.: Ecosystem Carbon Stock Assessment in Upland Forest: A Case Study in Koh Kong, Mondulkiri, Preah Vihear, and Siem Reap Provinces, Environmental and Rural Development, 61, 2023.

Pan, Y., Birdsey, R. A., Phillips, O. L., Houghton, R. A., Fang, J., Kauppi, P. E., Keith, H., Kurz, W. A., Ito, A., Lewis, S. L., Nabuurs, G.-J., Shvidenko, A., Hashimoto, S., Lerink, B., Schepaschenko, D., Castanho, A., and Murdiyarso, D.: The Enduring World Forest Carbon Sink, Nature, 631, 563-569, https://doi.org/10.1038/s41586-024-07602-x, 2024.

Pan, Y. D., Birdsey, R. A., Phillips, O. L., and Jackson, R. B.: The Structure, Distribution, and Biomass of the World's Forests, Annual Review of Ecology, Evolution, and Systematics, Vol 44, 44, 593-+, https://doi.org/10.1146/annurev-ecolsys-110512-135914, 2013.

Staudhammer, C. L., Escobedo, F. J., and Blood, A.: Assessing Methods for Comparing Species Diversity from Disparate Data Sources: The Case of Urban and Peri-Urban Forests, Ecosphere, 9, e02450, https://doi.org/10.1002/ecs2.2450, 2018.

**Response to Referee's comments (RC2)**

**Manuscript DOI:** https://doi.org/10.5194/egusphere-2024-3784
**Comment's citation:** https://doi.org/10.5194/egusphere-2024-3784-RC2        posted on 20250519

**Comment 1:** Sovann and others describe a unique forest dataset from Cambodia. The measurements are largely interesting and certainly novel, but as presented I was left wondering why scientific questions were not asked and/or why hypotheses were not addressed? As noted, it made me wonder if the manuscript fits the scope for BG or if it's better placed in ESSD? Having read the paper however I'm convinced that it would be relatively straightforward to focus more strongly on scientific aspects of the dataset that would make the analysis appropriate for BG.

**Response:** We appreciate the reviewer's comments. We agree that a clearer articulation of scientific questions and hypotheses strengthens the manuscript's alignment with Biogeosciences' scope. We have therefore substantially revised the manuscript to frame it around explicit scientific objectives and testable hypotheses regarding the impacts of land-cover change on ecosystem characteristics. Specifically:

- We revised the research objectives to emphasize analytical aims and formulated hypotheses. We now explicitly hypothesize that land-cover change reduces stand structure, species diversity, and leaf functional traits, and marked changes in soil conditions. Additionally, we evaluate whether locally calibrated DBH–height relationships significantly improve AGB estimates compared to generalized models. (Lines 86–96)
- The Abstract has been revised to emphasize the analytical objectives rather than focusing on dataset description (Lines 15–18), and references to data-sharing links have been removed.
- The Introduction section was comprehensively rewritten to support the updated research objectives and hypotheses. (Lines 45–85)
- We revised each Results subsection to align with our revised objective, which emphasizes how land-cover change alters key ecosystem characteristics. Specifically, we revised the following sections:
  - Soil conditions (Lines 256–261)
  - Species diversity (Lines 277–289)
  - Leaf traits (Lines 311–317)
  - Stand structure attributes: DBH and tree height (Lines 320–327), Aboveground and deadwood biomass (Lines 329–332), Stem density and basal area (Lines 334–339, Fig. 3)
- We introduced a new main section, "Estimated Aboveground Biomass Based on DBH-Height Relationship," in both the Results and Discussion to support our second hypothesis of applying locally calibrated DBH–H relationships for improving aboveground biomass estimation. (Lines 371–391, 564)
- We also updated the Discussion sections to support the revised results above: section 4.2 species diversity (lines 437–442), 4.3 Leaf functional traits, 4.4 Stand structure attributes: DBH and tree height (Lines 488–495), Aboveground and deadwood biomass (lines 500–511)

**Comment 2:** 45: This statement may be a bit of an oversell…quite a lot is known about the deleterious impacts of clear-cutting primary tropical forests. To me, this is the opportunity to direct the reader to the particular questions that this study will address.

**Response:** Thank you for pointing this out. We agree and have revised the sentence to more accurately reflect the scope of existing literature and clarify our research objective. The revised sentence now reads: "While the ecological consequences of forest conversion are broadly recognized, relatively few studies have comprehensively examined how transitions from primary to secondary forests and plantations influence multiple ecosystem characteristics, particularly through detailed field-based observations that may inform our understanding of ecosystem functioning." (Lines 45–48)

**Comment 3:** The introduction makes me wonder if the manuscript in general is more appropriate for ESSD given the focus on data. But in my opinion the authors can have it both ways…if they simultaneously describe the dataset and its major findings as it relates to a question or hypothesis clearly articulated in the introduction (as one would expect), the manuscript would in my opinion be more in scope with the journal to discuss interactions between biological, chemical, and physical processes. One idea would be to ask questions about and focus more of the results and discussion about the interesting variation between AGB methods where AGB_wd gives a far greater CP biomass than the other methods. The statistics in describing this in Figure 4b was rather weak (e.g. no p-value on page 332, what test was used?). Strengthening this will make the important methodological study more scientifically rigorous. In my opinion, the science can focus on important methodological challenges and the

meteorological instrumentation and discussion of other measurements wouldn't be a distraction because these things are partly responsible for forest growth in the first place.

**Response:** Thank you for your suggestions. We agree that stronger integration of our dataset descriptions with clearly articulated scientific questions and hypotheses is essential to position the manuscript within the scope of Biogeosciences, which is strongly related to forest biodiversity and ecosystem function. In response:

- We substantially revised the Introduction to focus on research questions and hypotheses that assess (i) how land-cover conversion alters ecosystem characteristics and (ii) the implications of using different allometric approaches for AGB estimation (Lines 86–96, See comment response 1).
- We strengthened the Discussion of the notable AGBwd overestimation in cashew plantations (CP). We now clarify that this is primarily driven by the wood density value (0.45 g cm⁻³) used in AGBwd, compared to 0.18 g cm⁻³ in the original AGBf function. This likely reflects differences in cashew wood properties or in the wood density measurement protocols used between the two studies. (Lines 598–600)
- Regarding the statistical analysis in Figure 4b, despite clear differences in mean AGB estimates among the methods, formal statistical comparisons were not conducted due to the limited number of plots per class (n = 3), which restricts statistical power. We now explicitly acknowledge this limitation in the Discussion. (Lines 601–602)
- We appreciate your recognition of the meteorological data as valuable context for understanding forest growth, and we have retained this data description in the revised manuscript.

**Comment 4:** 4.3: noting in the intro the importance of these comparisons of biodiversity across different forests and asking questions or stating hypotheses to test would help move the manuscript from purely descriptive to more of a scientific study.

**Response:** Thank you. We have major revised the Introduction to include research objectives and hypothesis (See comment response 1).

**Comment 5:** Fig. 1: It is a bit difficult to see from the figure where the site is in Cambodia…making the subplot to the upper right focus more clearly on the Thailand/Cambodia/Vietnam region would help place a stronger focus on Cambodia's geography.

**Response:** Thank you for pointing this out. We have updated the upper-right subplot in Figure 1 to provide a closer view of the Thailand–Cambodia–Vietnam region, improving geographic context and emphasizing the location of the study site in Cambodia. We also used our detailed land cover map, recently published by Sovann et al. (2025), as the background layer. (Line 106)

**Comment 6:** 82: of the 13% forested area, how much is primary and how much is secondary?

**Response:** Thank you for your comments. We have revised the recently updated land cover statistics in Kulen based on our recently published paper as the following "In 2021, 72 % of Phnom Kulen National Park was forested, dominated by nearly intact tropical evergreen forests (EF) (30 %) and forests that regrow naturally after clear-cutting (RF) (7 %). The remaining 35 % of forest cover consisted of semi-evergreen, deciduous, and bamboo stands. Non-forest areas were dominated by household-scale cashew plantations (CP) (15 %), with the remaining 13% consisting of croplands, paddy fields, settlements, and tree and rubber plantations (Sovann et al., 2025)." (Lines 101–105)

**Comment 7:** The Discussion as a whole is relatively strong with lots of good comparisons from the literature, which only further makes the point that the study could be converted to a scientific analysis relatively easily.

**Response:** We sincerely appreciate your kind recognition of the Discussion and the comparative analysis provided. Please refer to our comment response 1.

**Comment 8:** Why were the differences in AGB values from the different methods not discussed in the discussion? The analysis of this relatively more important component of the work needs to be improved.

**Response:** Thank you for pointing this out. Please refer to our comment response 3.

**References**

Sovann, C., Olin, S., Mansourian, A., Sakhoeun, S., Prey, S., Kok, S., and Tagesson, T.: Importance of Spectral Information, Seasonality, and Topography on Land Cover Classification of Tropical Land Cover Mapping, https://doi.org/10.3390/rs17091551, 2025.

**Response to Referee's comments (RC3)**

**Manuscript DOI:** https://doi.org/10.5194/egusphere-2024-3784
**Comment's citation:** https://doi.org/10.5194/egusphere-2024-3784-RC3        posted on 20250519.

**Comment 1:** The tropical forest ecosystem represents a vital ecological entity that sustains unparalleled biodiversity while delivering essential ecosystem services. This study provides a groundbreaking contribution through its comprehensive dataset encompassing multiple ecosystem characteristics - the first of its kind for this ecologically sensitive region. The wealth of empirical evidence collected from this biodiversity hotspot within a natural reserve renders the work scientifically valuable and worthy of publication consideration. However, the current manuscript adopts a descriptive cataloguing approach, which may limit its engagement value for readers. Furthermore, the structural organization requires refinement to enhance logical coherence, particularly through the development of a conceptual framework that systematically connects observed ecological patterns to their underlying processes. I suggest the authors to restructure the paper focusing on land-use/cover change impacts, and prioritize hypothesis-testing over pure data reporting. It would be better to tell a question-driven story rather than report a conventional data inventory.

**Response:** We appreciate the reviewer's comments. We agree that a clearer articulation of scientific questions and hypotheses strengthens the manuscript's alignment with Biogeosciences' scope. We have therefore substantially revised the manuscript to frame it around explicit scientific objectives and testable hypotheses regarding the impacts of land-cover change on ecosystem characteristics. Specifically:

- We revised the research objectives to emphasize analytical aims and formulated hypotheses. We now explicitly hypothesize that land-cover change reduces stand structure, species diversity, and leaf functional traits, and marked changes in soil conditions. Additionally, we evaluate whether locally calibrated DBH–height relationships significantly improve AGB estimates compared to generalized models. (Lines 86–96)
- The Abstract has been revised to emphasize the analytical objectives rather than focusing on dataset description (Lines 15–18), and references to data-sharing links have been removed.
- The Introduction section was comprehensively rewritten to support the updated research objectives and hypotheses. (Lines 45–85)
- We revised each Results subsection to align with our revised objective, which emphasizes how land-cover change alters key ecosystem characteristics. Specifically, we revised the following sections:
    - Soil conditions (Lines 256–261)
    - Species diversity (Lines 277–289)
    - Leaf traits (Lines 311–317)
    - Stand structure attributes: DBH and tree height (Lines 320–327), Aboveground and deadwood biomass (Lines 329–332), Stem density and basal area (Lines 334–339, Fig. 3)
- We introduced a new main section, "Estimated Aboveground Biomass Based on DBH-Height Relationship," in both the Results and Discussion to support our second hypothesis of applying locally calibrated DBH–H relationships for improving aboveground biomass estimation. (Lines 371–391, 564)
- We also updated the Discussion sections to support the revised results above: section 4.2 species diversity (lines 437–442), 4.3 Leaf functional traits, 4.4 Stand structure attributes: DBH and tree height (Lines 488–495), Aboveground and deadwood biomass (lines 500–511)

**Comment 2:** This title sounds too general, "Characteristics of ecosystems under various anthropogenic impacts in a tropical forest region of Southeast Asia".

**Response:** Thank you for pointing this out. In response, we revised the title to more clearly reflect the specific ecosystem characteristics examined, the land-use change context, and geographic focus. The new title is: "Land-cover change alters stand structure, species diversity, leaf functional traits, and soil conditions in Cambodian tropical forests". (Line 1)

**Comment 3:** At line 10 in the abstract: "anthropogenic pressure", which specific pressure? I would rather point out the importance of land-use/land cover change.

**Response:** Thank you for pointing this out. We have replaced "anthropogenic pressure" with the more specific term "land-use and land-cover change" to improve clarity and accurately reflect the main driver examined in this study. (Line 10)

**Comment 4:** At line 11, "both in pristine tropical forests and in the converted deforested" As you mentioned, what was done in this research, land-cover change may be the main focus?

**Response:** Thank you for your comment. We agree that land-cover change impact on ecosystem characteristics is the main focus of this study. To clarify this, we revised the manuscript as follows:

- Updated the abstract (Line 10) and introduction (Lines 86–96) to clearly state the research questions and hypotheses, explicitly highlighting land-cover conversion as the core driver of ecosystem change;
- Revised the title to reflect this emphasis (See our response to Comment 2);
- Reframed the manuscript around explicit scientific objectives and testable hypotheses concerning the impacts of land-cover change on ecosystem characteristics (See our response to Comment 1).

**Comment 5:** At line 20, suggested replacing "filling data gaps" with "enriching databank".

**Response:** We agree with the suggestion and have revised the sentence as recommended. (Line 22)

**Comment 6:** At line 21, suggested deleting "addressing global environmental challenges".

**Response:** We accepted the suggestion and deleted "addressing global environmental challenges". (Line 22)

**Comment 7:** At line 36, as mentioned, "resulting in a decrease in biodiversity", suggested adding "The last sentence of a paragraph should call for your research focus/significance, which is not "biodiversity" I guess?"

**Response:** Thank you for pointing this out. In response, we revised the sentence from "resulting in a decrease in biodiversity" to "Such disturbances have resulted in significant structural and functional degradation in tropical forests, highlighting the urgent need to assess how land-cover change alters key ecosystem characteristics." This revision more accurately reflects the study's focus on the impacts of land-cover change on ecosystem characteristics. (Line 35–37)

**Comment 8:** At line 46, suggested deleting "biodiversity and" for clarity.

**Response:** We agree, and the sentence has been revised (Line 45–48, also see our response to Comment 2 from Reviewer 2).

**Comment 9:** At line 47, suggested replacing "environmental challenges" with "land cover?" and adding "it is necessary to conduct…" to the sentence. Also recommended revising "data are necessary to assess the dynamic…" to "in order to investigate the…".

**Response:** Thank you for pointing this out. We agree with your comment and have revised the manuscript accordingly (Lines 60–61).

**Comment 10:** At lines 59-62, suggested deleting the sentence in the lines and writing sth about the importance of your study region/object, show the readers why you do studies here.

**Response:** Thank you for the suggestion. We have deleted the original sentence at lines 60–63 and added a brief explanation of the importance of our study region, adapted from the Study Area section. This change clarifies why Phnom Kulen National Park was selected and now appears at lines 77–85 of the revised manuscript.

**Comment 11:** At line 65, "ecosystem level", are you sure? some also at plot level?

**Response:** We revised the research objectives to emphasize analytical aims and explicitly formulated hypotheses; therefore, the original sentence at line 65 was deleted. (Lines 86–96)

**Comment 12:** Can you come up with specific scientific questions and also give hypotheses?

**Response:** See our response to Comment 1.

**Comment 13:** At lines 74-81 in the section "2.1 Study area and selection of plots", move the sentences at the lines to the introduction.

**Response:** We agree, and have moved the sentence to the introduction as noted in our response to Comment 9.

**Comment 14:** At line 295 in section "3.4.1 DBH-H relationship", no need to discuss every result, just focus on your key findings.
**Response:** Thank you for the suggestion. We revised the DBH-H relationship section (Lines 320–327) to highlight only the key findings relevant to structural changes among land-cover types. We streamlined the presentation by removing excessive detail and focusing on the main trends in DBH and height, aligning the section more closely with our revised hypotheses.

**Comment 15:** At line 378, there are too many subsections in the section "4 Discussions". I would suggest going with three major aspects: Meteo, plant, and soil. Also, maybe discuss these aspects linking with your main scientific questions (land-cover change)?
**Response:** Thank you for your comments. In response, we substantially revised both Results and Discussion sections to better reflect our revised research objectives and hypotheses. Specifically:
- We removed Section 4.1 ("Importance of tropical field data"), integrating relevant content into the Introduction to improve narrative focus. (Line 389, 48–63)
- To address the excessive sub-sectioning in the Discussion section, we simplified the structure by:
  - Removing subsection numbering under the result section "3.4 Stand structure attributes" and discussion section "4.4 Stand structure attributes" (Lines 319, 328, 333, 347, 487, 499, 512, 538)
  - Merging "LAI and fPAR" section to be a subsection of "Stand structure attributes" for both results and discussion sections. (Lines 347, 538)
- We restructure the paragraphs in both Results and Discussions to focus on the revised hypotheses and remove less relevant information from the manuscript (See our response to Comment 1).

**Comment 16:** At line 379, in section "4.1 Importance of tropical field data", no need to discuss here; you already told this in the introduction.
**Response:** Thank you for pointing this out. We removed Section 4.1 ("Importance of tropical field data"), integrating relevant content into the Introduction to improve narrative focus. (Line 410, 60–76)

**Comment 17:** At line 576, in section "5 Conclusions", looks like a summary rather than conclusions, and it's too long. Here it's better to provide your key findings, further instructions, or more practical suggestions for the future. At lines 577-585, shorten the sentences at these lines into one sentence showing the background of this work.
**Response:** Thank you for your suggestions. In response, we revised the conclusion to emphasize key findings and research gaps for future studies, rather than repeating a summary of results. Specifically:
- We condensed the original background sentences (Lines 577–585) into a single sentence summarizing the study's rationale and objectives. (Lines 622–625)
- We added recommendations for future research, such as conducting destructive sampling to validate locally calibrated AGB equations and expanding field data collection across a broader range of land-cover types in Southeast Asia. (Lines 634–638)